# Temporal Concept Dynamics in Diffusion Models via Prompt-Conditioned Interventions

**Ada Görgün**[1,*]    **Fawaz Sammani**[2,*]    **Nikos Deligiannis**[2]    **Bernt Schiele**[1]    **Jonas Fischer**[1]

[1]Max Planck Institute for Informatics, Saarland Informatics Campus, Germany
[2]ETRO Department, Vrije Universiteit Brussel and imec, Belgium
`{agoerguen, schiele, jonas.fischer}@mpi-inf.mpg.de`
`{fawaz.sammani, nikos.deligiannis}@vub.be`
∗ Equal contribution.

## Abstract

Diffusion models are usually evaluated by their final outputs, gradually denoising random noise into meaningful images. Yet, generation unfolds along a trajectory, and analyzing this dynamic process is crucial for understanding how controllable, reliable, and predictable these models are in terms of their success/failure modes. In this work, we ask the question: *when* does noise turn into a specific concept (e.g., age) and lock in the denoising trajectory? We propose PCI (Prompt-Conditioned Intervention) to study this question. PCI is a training-free and model-agnostic framework for analyzing concept dynamics through diffusion time. The central idea is the analysis of *Concept Insertion Success* (CIS), defined as the probability that a concept inserted at a given timestep is preserved and reflected in the final image, offering a way to characterize the temporal dynamics of concept formation. Applied to several state-of-the-art text-to-image diffusion models and a broad taxonomy of concepts, PCI reveals diverse temporal behaviors across diffusion models, in which certain phases of the trajectory are more favorable to specific concepts even within the same concept type. These findings also provide actionable insights for text-driven image editing, highlighting *when* interventions are most effective without requiring access to model internals or training, and yielding quantitatively stronger edits that achieve a balance of semantic accuracy and content preservation than strong baselines. Code is available at: PCI Framework

## 1 Introduction

Text-to-Image diffusion models have emerged as a powerful class of generative frameworks Rombach et al. (2022a); Podell et al. (2024); Esser et al. (2024a), redefining the landscape of text-driven image synthesis and editing. By progressively reducing random noise to structured output, these models capture rich data distributions and enable realistic, creative, and novel image synthesis. Despite their success, the internal dynamics of how semantic concepts (e.g., gender, style, ethnicity) emerge and solidify during the denoising trajectory remain poorly understood. For more targeted steering and intervention of the generation, and better understanding of potential sources of systematic differences within the model, getting a better understanding on how these dynamics look is essential. Without such understanding, attempts to debug, guide, or control generation remain largely heuristic.

Existing methods usually investigate concepts in diffusion models through the lens of attribution maps (Tang et al., 2022; Helbling et al., 2025), which localize concepts in generated images and answer *where* but not *when* concepts manifest over time. Other works explore concepts in diffusion models using Concept Bottleneck Models Ismail et al. (2024); Kulkarni et al. (2025) or Sparse Autoencoders Cywiński & Deja (2025), which map intermediate latent noise to human-interpretable concepts. These approaches, however, often overlook temporal dynamics, involve complex (and sometimes supervised) concept-naming procedures, are not faithful to the original model with usually deteriorating performance, and require extra training. They are also largely confined to simple datasets and domains, limiting their scalability to the typical real-world settings.

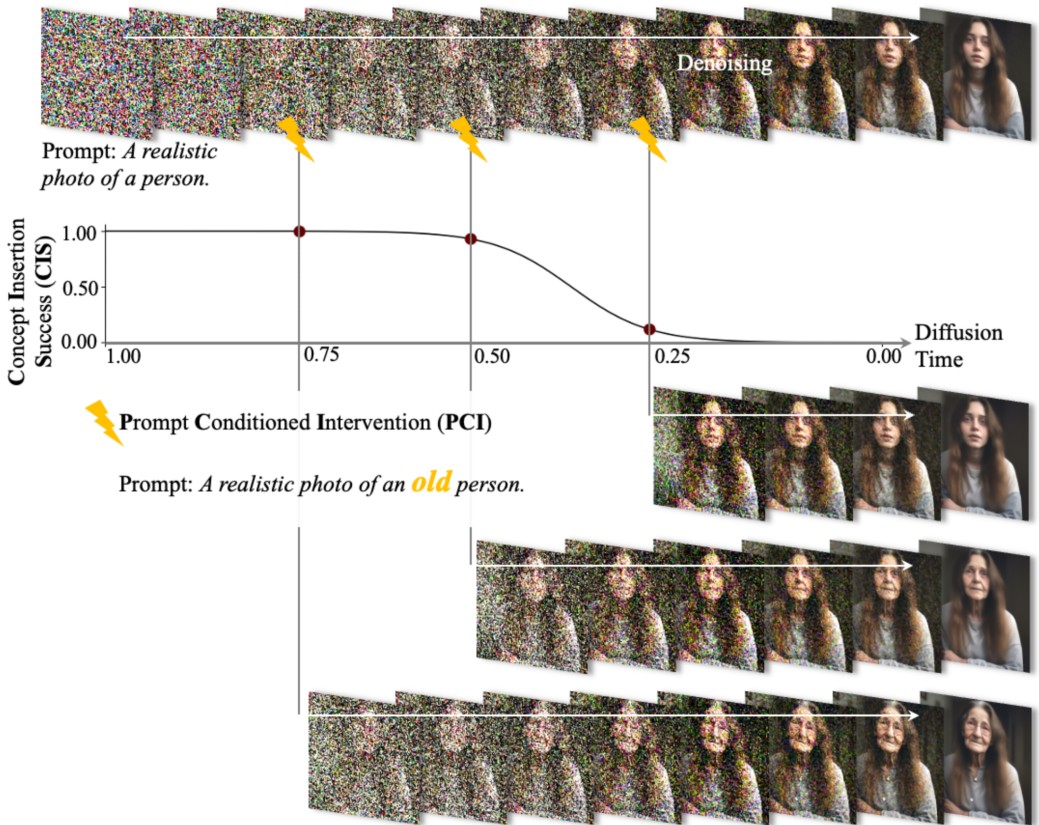

Figure 1: **Prompt-conditioned intervention (PCI) over diffusion timesteps.** We suggest to study *when* noise turns into a specific concept through the lens of concept insertion success, i.e., the chance of inserting a concept at a certain timestep successfully. By switching the base-prompt (top) at different time points of the diffusion process with a prompt composed of the base prompt *and* the concept of interest (old), we can measure this success rate (CIS curve) across different seeds and base prompts to analyze temporal dependency and influence of concepts in diffusion models.

In this work, we shed light on the temporal evolution of concepts in diffusion and flow matching models by introducing Prompt-Conditioned Interventions (PCI), a lightweight and model-agnostic mechanism in which we switch the text prompt at a chosen timestep and observe how the model reacts. This probing procedure allows us to quantify when a concept can still influence the trajectory and when the model has already committed to a semantic interpretation and investigate a wide range of temporal behaviors: *concept insertion*, by measuring when a new attribute can still be added; *concept deletion*, by testing when an existing attribute can be overridden; and *multi-concept interactions*, by examining how two or more attributes jointly affect one another when introduced at different times. Within this broader probing framework, our primary quantitative measure is the **Concept Insertion Success (CIS)** curve, which captures the probability that a concept remains insertable at each timestep.

Concretely, consider the example in Fig. 1. Starting from the base prompt *"a realistic photo of a person"*, an image is generated through the denoising process of diffusion models. Prompt-Conditioned Interventions (PCI) intervene (⚡) at different timesteps, augmenting the base prompt with a target concept, here *"a realistic photo of an old person"*. Generation then continues with this augmented prompt as prior. If the resulting image incorporates the target concept old, the intervention timestep marks a point at which the concept can still be flexibly and successfully inserted (high CIS). If intervening yields no meaningful change and produces an image nearly identical to the one from the base prompt, the concept is effectively locked and no longer influences the trajectory (low CIS). PCI allows us to derive CIS curves for any concept, in any context, and for any diffusion- or

flow-based model, offering a practical tool for understanding their temporal behavior. By analyzing 800 samples spanning diverse categories and fine-grained concepts across five generative models, we observe the following trends: **(1) Timing matters:** global factors (time, weather, season, color) lock in early; human attributes (e.g., age, gender) stabilize in the middle; and out-of-distribution concepts (e.g., "horse in living room") insert unusually early. **(2) Model choice matters:** rectified-flow models insert concepts earlier, while diffusion models preserve editability deeper into the trajectory. **(3) Context matters:** alignment between concept and prompt context strongly shapes insertability, small context shifts (e.g., outdoor→indoor) can flip a concept from late-insertable to early-insertable, while seed noise is largely suppressed by averaging.

Beyond analysis, PCI also enables a practical, training-free editing application. Because CIS identifies when a concept can be modified without disrupting unrelated content, it provides a principled way to select effective editing windows. By switching the prompt at a CIS-informed timestep, we obtain edits that introduce the desired concept while staying close to the original trajectory—without relying on attention maps Epstein et al. (2023); Hertz et al. (2023) or segmentation modules Tewel et al. (2025). Quantitatively, this CIS-guided editing achieves a superior balance between preservation and semantic strength compared to strong baselines such as NTI+P2P Mokady et al. (2023) and Stable Flow Avrahami et al. (2025).

In summary, our contributions are as follows:

- We introduce PCI, a unified temporal probing mechanism for text-to-image models. Applied to the insertion setting, it yields the Concept Insertion Success (CIS) curve, quantifying the probability that a concept is insertable at each timestep.
- We analyze a wide range of concepts, contexts, and models, including multi-concept interactions and concept deletion, revealing consistent temporal patterns of concept emergence and stabilization.
- We demonstrate a practical editing application enabled by CIS, showing that timing materially affects edit reliability and fidelity, and that CIS-guided intervention windows offer actionable guidance on *when* to apply edits for high-quality, content-preserving results.

## 2 RELATED WORK

**Diffusion Models** (Ho et al., 2020; Dhariwal & Nichol, 2021) form the foundation of modern generative imaging, widely applied in both image (Rombach et al., 2022b; Saharia et al., 2022) and video generation (Esser et al., 2023). While early approaches rely on stochastic differential equations (SDEs) (Song et al., 2021), more recent formulations such as Rectified Flow (Liu et al., 2023) and Flow Matching (Lipman et al., 2023) through ordinary differential equations (ODEs) (Albergo & Vanden-Eijnden, 2023) provide faster and more stable training. At the same time, architectures have evolved from UNets (Ho et al., 2020) to Diffusion Transformers (DiT) (Labs, 2024; Esser et al., 2024b; Sauer et al., 2024), enabling greater scalability (Peebles & Xie, 2023). Yet, this progress has outpaced our ability to understand their reasoning and generation dynamics, which is however essential for trust, reliability, and safe deployment (Shi et al., 2025).

**Static Interpretability** in text-to-image diffusion models examine how textual concepts manifest in the generated images. Recent attribution-based approaches localize where prompt tokens appear in the image (Tang et al., 2022; Helbling et al., 2025). Such approaches provide spatial grounding that supports faithfulness checks, safety auditing, and region-level controllability. Other methods instead map internal latents to human-interprepretable concepts. Here, concept-bottleneck layers or sparse autoencoders are learned either aligning latent directions with predefined attributes (Ismail et al., 2024; Cywiński & Deja, 2025) or fitting to large-scale text embeddings (e.g., CLIP) (Kulkarni et al., 2025; Kim & Ghadiyaram, 2025). An alternative approach to learn such mappings are pseudo-tokens in the text encoder that represent a concept as a weighted combination of vocabulary embeddings (Chefer et al., 2024). These approaches clarify *what* the model represents and *where* in the image it appears. However, they are typically evaluated only at a single late stage of generation, treating the model as a *static mapping*, providing only limited insight into the model's *temporal dynamics*. The questions *when* and *how* concepts are insertable or controllable remain unanswered.

**Dynamic Interpretability** approaches in contrast recognize that a diffusion model's behavior changes throughout the denoising time. Wu et al. (2022) discovered that Stable Diffusion models naturally disentangle high-level content from stylistic attributes over time. Similarly, Chen et al. (2024b) show

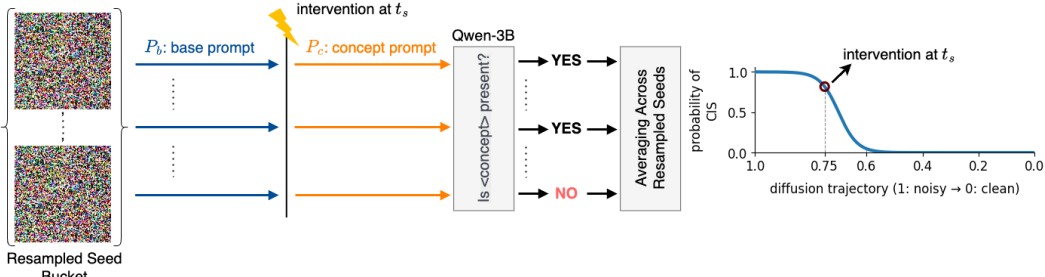

Figure 2: **Overview of the PCI framework.** A base prompt $P_b$ is used as conditioning for generation, altered to the concept prompt $P_c$ at time $t_s$. The generated images are evaluated through VQA to determine concept presence and aggregated across seeds to obtain CIS across the diffusion trajectory.

that diffusion-based editing is highly sensitive to the choice of timestep and noise initialization. By training multiple sparse autoencoders at different timesteps in the trajectory, researchers revealed early formation of scene layout and mid-trajectory fixation of object composition, yet without enforcing concept consistency across time (Tinaz et al., 2025). Huang et al. (2025) instead train a single autoencoder across the entire trajectory to encourage temporal coherence of the learned representations, but do not aim for dynamic concept-level interpretability. Unlike timestep-sensitive editing methods (Wu et al., 2022; Chen et al., 2024b) or autoencoder-based probes (Tinaz et al., 2025; Huang et al., 2025), PCI is a training- and annotation-free approach to investigate when a diffusion model commits to a concept directly from the model's generative behavior instead of surrogates.

## 3 METHODOLOGY

In this section, we present our method for analyzing the temporal evolution of concepts in diffusion models. We refer to Appendix Sec. A.1 for preliminaries on diffusion models. At a high level, our proposed method PCI (Sec. 3.1) begins with a base prompt that guides the generation up to a specific timestep. At that timestep, we replace the base prompt with an augmented version that combines the base prompt with the target concept we wish to analyze. The generation then continues with this augmented prompt and an image is synthesized. Finally, a Large Vision Language Model (LVLM) with the task of Visual Question Answering (VQA) is used to detect the presence of the target concept in the synthesized image. By performing this intervention at every timestep, we obtain a Concept Insertion Success curve (Sec. 3.2), which quantifies, for each timestep, the probability that the concept is successfully inserted into the generation trajectory and expressed in the final image. A high-level overview of our framework is illustrated in Fig. 2. To complement CIS, we also explore **Concept Deletion Success (CDS)**, which mirrors this procedure but switches from a concept-bearing prompt back to the base prompt, quantifying when a concept can still be removed. While CDS provides a dual perspective on temporal concept behaviour indicating when a concept becomes difficult to erase, we do not adopt it as a primary metric as we found that it introduces confounding factors that make CDS less stable and interpretable across concepts and models. For this reason, CIS remains our main analytical tool and we refer to Appendix Sec. A.4 for further details regarding CDS.

### 3.1 PROMPT-CONDITIONED INTERVENTION (PCI)

Building on the iterative denoising framework of diffusion models, our method uses *Prompt-Conditioned Intervention* (PCI) to analyze the temporal evolution of semantic concepts, and the point where they become locked in the denoising trajectory and after which they do not influence the trajectory anymore. The key idea is to alter the denoising trajectory in an intermediate timestep $t_s$ by changing a part of the conditioning input (changing the text prompt to include a *concept*), and then continue the denoising process with this new condition to obtain a reconstructed image. This modification serves to isolate the effect of introducing a specific concept, allowing its influence on the generative trajectory to be explicitly traced. By comparing reconstructions across different timesteps and conditioning variants, PCI reveals how individual concepts influence the generation trajectory and at which stages they become visually embedded.

Formally, let $P_b$ denote the *base prompt*, a neutral description without the target concept (*e.g., a realistic photo of a person*). Let $P_c$ denote the *concept prompt*, obtained by augmenting $P_b$ with the explicit inclusion of the target concept $c$ (*e.g., a realistic photo of an old person*). We define $\text{Denoise}(x_t, y)$ as the denoising process of a diffusion model at timestep $t$, where $x_t$ is the noisy latent and $y$ is the text condition. Over timesteps $t \in \{T, \ldots, 0\}$, this process reconstructs a clean latent $x_0$. The synthesized image is obtained by decoding the clean (denoised) latent $\mathbf{x}_0$ through the decoder of the latent diffusion model. Given an initial noisy latent $\mathbf{x}_T$ and a conditioning text input $P_b$, we first ensure that the initial noisy latent $\mathbf{x}_T$ does not encode noise features that would lead to the generation of a latent image $\mathbf{x}_0$ that contains the target concept $c$. To perform this, we apply a seed resampling operation. The details of this procedure are provided in Appendix Sec. A.2. We start the denoising process with:

$$\mathbf{x}_{t_s} = \text{Denoise}(\mathbf{x}_T, P_b), \tag{1}$$

where $\mathbf{x}_{t_s}$ is the intermediate state reached under the text condition $P_b$ at timestep $t_s$. We then switch the conditioning from $P_b$ to $P_c$ at a timestep $t_s$:

$$\mathbf{x}_0\big(P_b \xrightarrow{t_s} P_c\big) = \text{Denoise}(\mathbf{x}_{t_s}, P_c), \tag{2}$$

As illustrated in Fig. 1, this formulation provides a mechanism for probing the sensitivity of the generation process to concepts and prompt alterations at different timesteps when prompt switching is performed at $t_s$: $P_b \xrightarrow{t_s} P_c$.

## 3.2 CONCEPT INSERTION SUCCESS (CIS)

A central objective of PCI is to determine *when* a concept becomes locked in the denoising trajectory, and after which it does not influence the trajectory anymore. We formalize this through **Concept Insertion Success (CIS)**, which is defined as the probability that a concept inserted through PCI at that timestep appears in the generated output. Intuitively, with CIS, we can investigate the earliest timestep where switching the conditioning from a base prompt $P_b$ to a concept prompt $P_c$ no longer alters the base generation obtained solely from $P_b$. By repeatedly applying the switch from $P_b$ to $P_c$ at varying points $t_s$ along the trajectory, where $s$ varies from $T \rightarrow 1$, we can measure when the introduced concept appears in the generated image. To efficiently quantify concept presence over time, we leverage Large Vision-Language Models (LVLMs) by posing the problem as Visual Question Answering (VQA). For each reconstruction obtained by prompt switching $\mathbf{x}_0(P_b \xrightarrow{t_s} P_c)$, the VQA model evaluates whether the concept $c$ is present by returning yes or no. While perceptual similarity (e.g., CLIPScore Hessel et al. (2021) or LPIPS Zhang et al. (2018)) would be an alternative, they proved less powerful and unspecific in preliminary experiments and having particular difficulties identifying *the absence* of a concept. We provide details on prompts in Appendix Sec. A.3.

## 4 EXPERIMENTS

We investigate when in a diffusion trajectory, it becomes unlikely that the underlying model can still introduce a new target concept into the generation process. Using our PCI approach (Sec. 3.1) we investigate the temporal dynamics in state-of-the-art diffusion models including Stable Diffusion 2.1 (SD 2.1) Rombach et al. (2022a), Stable Diffusion XL (SDXL) Podell et al. (2024), Stable Diffusion 3.5 (SD 3.5) Esser et al. (2024b); Sauer et al. (2024), PixArt-alpha XL Chen et al. (2024a) (a diffusion-based transformer (DiT) model) and FLUX.1-dev Labs (2024) for a large set of concept categories (Sec. 4.1) with our CIS scorer selected as Qwen-VL-3B (Bai et al., 2025) (see Appendix Sec. C.1 for ablations on multiple VQA models). All CIS measurements reported in this work are averaged across multiple random seeds (see details in Appendix Sec. C.2) to ensure robustness and reduce variance. We further verify stability under prompt-wording changes through our prompt-variation robustness test in Appendix Sec. C.3. Through our CIS metric (Sec. 3.2) we reveal differences in when certain types of concepts remain insertable into image generation as well as differences across models (Sec. 4.2) and show that even for the same concept class, depending on the context, there can be stark differences in when a concept can still be insertible (e.g., 'cat' versus 'horse') in Sec. 4.2. We provide all details on setup, evaluation protocol across models, and analysis metrics in Appendix Secs. B and B.2, respectively.

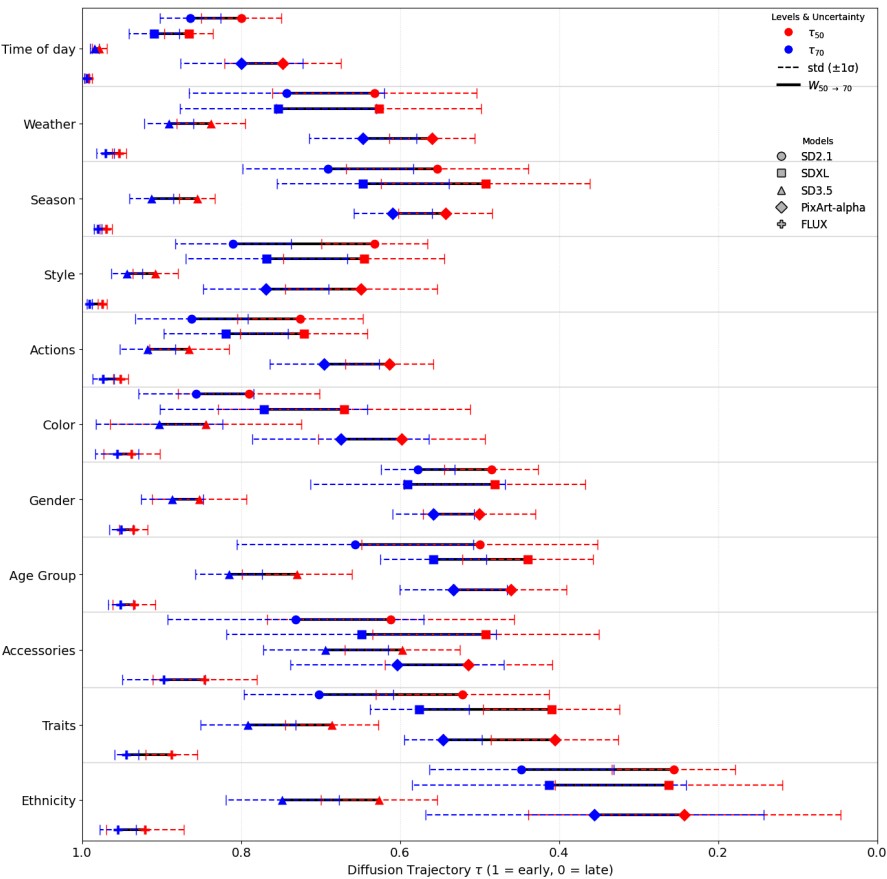

Figure 3: **CIS reveals cross-concept and cross-architecture differences.** CIS for ● $\tau_{50}$ and ● $\tau_{70}$ across multiple concept categories and diffusion models.

## 4.1 CONCEPT GROUPS

We consider a large set of concepts organized into a hierarchical structure of categories, subcategories, and individual concepts drawing inspiration from existing works in the literature Bakr et al. (2023); Huang et al. (2023). It covers demographics (e.g., gender, ethnicity, age group), objects (animals, handmade objects, natural elements), human properties (clothing, accessories, physical traits), actions (common human activities), attributes (material, color), environmental factors (season, weather, time of day, setting/venue), and style (sketch, painting, realistic). Each subcategory contains a diverse set of fine-grained concepts (e.g., for the age group, concepts are "baby", "child", "teenager", "adult", "old"), allowing systematic evaluation of models across a wide spectrum of concepts. Each concept is further represented across 8 different contexts (e.g., river in a landscape and river in a street), resulting in approximately 800 fine-grained concept descriptions in total. We refer to Appendix Sec. A.3 regarding how we designed the questions for the VQA model for each subcategory.

## 4.2 TEMPORAL CONCEPT INSERTABILITY

We quantify when concepts are reliably insertable along the denoising trajectory using band-limited CIS statistics computed *exclusively* within the CIS probability band [0.50, 0.70]. Our goal is twofold: (i) characterize *when* insertion is possible (crossing times) at the specified CIS bands, which we refer to as $(\tau_{50}, \tau_{70})$, and *how sensitive* it is to the exact intervention point (steepness) through band-width $W_{70\to50} = |\tau_{70} - \tau_{50}|$ that tracks the maximum slope on the CIS curves; and (ii) compare these behaviors across models, subcategories, and for more fine-grained intuition, across concept-prompt pairs. We refer to more detailed definitions in Appendix Secs. B and B.2. Beyond the quantitative CIS analysis, we include a complementary *qualitative* study of cross-attention maps (Appendix Sec. D.2) to visualize how concept strength and spatial focus evolve with the intervention timestep. We

additionally evaluate *multi-concept interactions* (Appendix Sec. D.3) and extend our analysis to *abstract concepts* (Appendix Sec. D.4), examining how these settings modify insertion dynamics.

CATEGORY-LEVEL ANALYSIS

In Fig. 3 we report category-level findings over the CIS metrics $\tau_{50}$ and $\tau_{70}$ across models, grouped into subcategories defined as in Sec. 4.1. Due to space limitations, we only include a subset of categories in Fig. 3 and defer the rest to Appendix Sec. D.

**Cross-model Trends within Subcategories.** Based on our analysis in Fig. 3, we observe a stable timing hierarchy and alignment rule across SD 2.1, SDXL, SD 3.5, PixArt-alpha, and FLUX.1-dev despite their architectural and scheduling differences:

- **Monotone CIS.** For all subcategories and models, $C(\tau)$ is empirically nondecreasing, yielding well-defined level-*q crossing times* $\tau_q$ (see Fig. 4a for SDXL).

- **Global factors emerge earliest and most sharply.** *Style*, *time of day*, *weather*, *season* and *color* exhibit large $\tau_q$ (they are inserted early) (Fig. 3) and narrow steep windows $W_{70 \to 50}$

- **Human properties sit in between.** While *gender* and *age group* are inserted mid-time; *ethnicity* tends to be inserted later (smaller $\tau_q$) than other demographics in SD 2.1/SDXL. Expressive cues (*traits*, *accessories*) follow the same ordering, with *accessories* typically having the latest insertion among them. (Fig. 3)

**Model-specific temporal patterns.** Based on our analysis in Fig. 3, we observe model specific patterns that allows us to perform a distinct comparison regarding how they treat concepts within the diffusion trajectory.

- **Rectified flow models.** Compared to diffusion-based models such as SD 2.1 (U-Net architecture), SDXL (U-Net architecture), and PixArt-alpha (DiT architecture), rectified-flow models such as SD 3.5 and FLUX.1-dev consistently *sharpen* the transition (smaller band $W$) across categories. In many cases this sharpening coincides with *earlier* crossings (larger $\tau_q$), reducing late-stage flexibility (Fig. 3). As a result, concepts in rectified flow models are inserted much earlier than in diffusion models.

- **Accessories as an exception.** While SD 3.5 generally enforces early insertion for human properties, *accessories* stand out: $\tau_{50}=0.53\pm0.05$ and $\tau_{70}=0.62\pm0.03$ place the crossing window near the mid-trajectory, substantially later than SD 3.5 *traits* ($\tau_{50}=0.69$). Cross-model comparison shows accessories are the most late-flexible human concept (SD 2.1: 0.46/0.58; SDXL: 0.41/0.53; SD 3.5: 0.53/0.62), though SD 3.5 still yields larger $\tau_q$ than SD 2.1/SDXL at both levels. Unlike other human concepts in SD 3.5, accessories can often be adjusted mid-trajectory without being washed out, offering rare late-stage flexibility within this model family.

- **SD 2.1 and late-stage flexibility.** SD 2.1 preserves more *late-stage flexibility* (smaller $\tau_q$ in multiple appearance categories) but at the cost of broader and more timing-sensitive windows (larger $W$) (Fig. 3).

**Key findings.** Our analysis isolates two drivers of *when* edits take effect and *how* reliably they succeed along the generation trajectory:

1. **Timing of concept insertion.** Global scene factors (e.g., style, time of day, weather, season and color) are inserted and locked in early; Human attributes (e.g., age, gender) typically lock in mid-trajectory, and fine-level details such as accessories typically lock in mid-to-late trajectory.

2. **Model choice.** Rectified-flow models insert concepts earlier than diffusion models, whereas diffusion models allow later insertions.

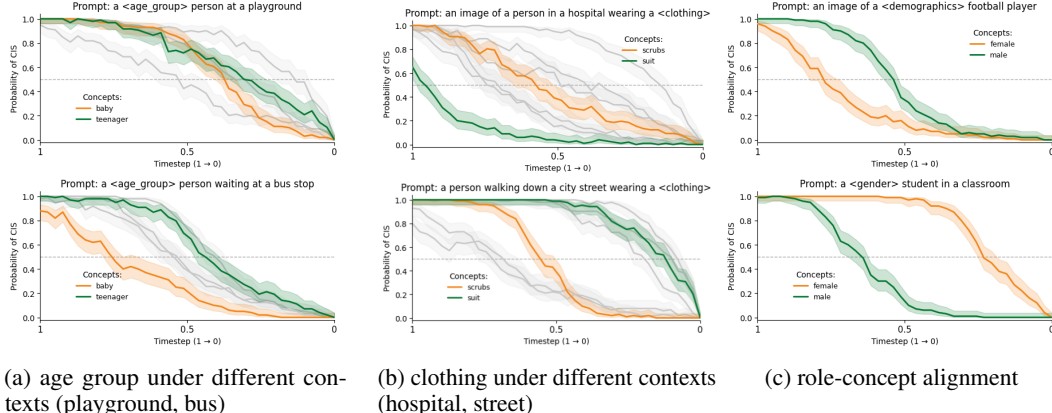

(a) age group under different contexts (playground, bus)

(b) clothing under different contexts (hospital, street)

(c) role-concept alignment

Figure 4: **Revealing context-dependent differences for the same concept.** We show the difference in CIS curves of a concept provided with different context in the base prompt.

FINE-GRAINED ANALYSIS

While category-level summaries highlight broad timing patterns, they can mask the specific prompts (contexts) and concepts that underlie these patterns. Fine-grained analyses investigates these sources of variation by showing full CIS curves at the level of (i) a fixed concept across different prompts/contexts (prompt-level), and (ii) different concepts within the same subcategory under matched prompts (concept-level). This reveals when the aggregated behavior of a category is dominated by a subset of contexts or a subset of its concepts and highlights *meaningful concept–context relationships* that are obscured by averaging (e.g., how the same concept behaves across distinct settings, or how different concepts within a category respond to the same context setting). For this, we carry out fine-grained analyses for SDXL. Detailed *prompt-level* and *concept-level* comparisons are provided in Appendix Sec. D.1, where we vary either the base prompt (context) or the concept within a subcategory and report CIS curves with **95% Wilson CIs** over seeds. Here, we highlight additional fine-grained findings at the concept-level that reveal nuanced differences within different concept-context interactions.

**Additional case studies.** In Fig. 4, we present several examples illustrating how concept–context relationships determine when a concept can be successfully introduced during the diffusion trajectory. These cases demonstrate that the alignment between a concept and its surrounding context plays a central role in shaping CIS. For instance, within the subcategory "age group" in public settings (Fig. 4a), a *baby* is inserted later in the trajectory of a "playground" than in that of a "bus stop" since the playground provides a more natural and supportive context, allowing for later insertions. By contrast, a *teenager* shows the opposite pattern: it is locked earlier in the playground trajectory than in the bus stop trajectory. A similar dynamic emerges when comparing clothing (scrubs, suit) with workplace contexts (hospital, street) in Fig. 4b. In the less typical context of "street" scrubs lock in earlier in the diffusion trajectory than they do in a "hospital" whereas the pattern reverses for a *suit*. The same phenomenon is visible in Fig. 4c where in uncommon scenarios such as a female football player or a male cashier, the gender category is inserted earlier in the diffusion trajectory, and the CIS window of persistence becomes narrower.

**Finding:** Out-of-distribution (OOD) concepts relative to context (e.g., *horse in a living room*, *wearing scrubs in a street*, *baby at a bus stop*) become locked in earlier than common in-distribution concepts relative to context (e.g., *cat in a living room*, *wearing scrubs in hospital*, *baby at a playground*). These OOD pairs inflate $\tau_q$ early and $W$ shows greater timing fragility.

## 5 TEXT-DRIVEN IMAGE EDITING

The CIS probability function derived from PCIs enables feature-preserving text-driven image editing, where a generated image is modified according to a text prompt while maintaining the original

| Methods | $\text{CLIP}_{\text{img}}$ | $\text{CLIP}_{\text{txt}}$ | $\text{CLIP}_{\text{dir}}$ |
|---|---|---|---|
| NTI+P2P | 0.8666 | 0.2215 | 0.0979 |
| Stable Flow [CVPR 2025] | 0.8324 | 0.2152 | 0.0631 |
| PCI-$\tau_{30}$ | 0.9343 | 0.2125 | 0.1014 |
| PCI-$\tau_{50}$ | **0.8885** | 0.2236 | 0.1387 |
| PCI-$\tau_{60}$ | 0.8625 | 0.2289 | 0.1531 |
| PCI-$\tau_{70}$ | 0.8353 | **0.2341** | **0.1678** |
| PCI-$\tau_{90}$ | 0.7679 | 0.2449 | 0.1963 |

Table 1: **Quantitative comparison of editing methods.** We report $\text{CLIP}_{img}$ (content preservation), $\text{CLIP}_{txt}$ (semantic alignment), and $\text{CLIP}_{dir}$ (directional consistency). Higher $\text{CLIP}_{img}$ indicates better preservation, while higher $\text{CLIP}_{txt}$ and $\text{CLIP}_{dir}$ indicate stronger concept insertion.

content as accurately as possible. Consider a concept category (e.g., age group) within a broader context, which we refer to as the base prompt (e.g., a person at a city park). We first compute the CIS function for each fine-grained concept of the concept category (e.g., baby, child, young, adult and old) and then average across them to obtain a representative CIS function for the subcategory. This function encodes the probability that any fine-grained concept associated with the subcategory can be successfully inserted at a given timestep.

To edit an image with a desired concept, we map user-defined probability to its nearest corresponding timestep $t_s$ on the CIS curve and run PCI. That is, we perform prompt switching at $t_s$, replacing the base prompt augmented with the new concept for the remainder of the generation. Fig. 5 illustrates text-driven image editing for SDXL, and we refer to Appendix Sec. D.5 (Fig. D6) for examples on SD 3.5. We show edits at different CIS probabilities $\{0.3, 0.5, 0.7, 0.9\}$. From the analysis in Sec. 4, we find that edits are most reliable within the interval $[\tau_{50}, \tau_{70}]$, which is also reflected in Fig. 5. High values at $\tau_{90}$ achieve the intended modification but compromise content preservation, while low values at $\tau_{30}$ do not affect the base image, as the concept no longer influences the trajectory.

To rigorously assess the effectiveness of CIS-guided editing, we conducted a quantitative comparison against two strong baselines: NTI+P2P Mokady et al. (2023) and Stable-Flow Avrahami et al. (2025). For this purpose, we constructed a dedicated evaluation dataset consisting of **88 concept pairs**, each comprising a base prompt and a corresponding concept prompt, and generated **20 random seeds per concept**, resulting in a total of **1760 edited images** across all methods. Following prior work Avrahami et al. (2025), we employ three complementary CLIP-derived metrics to measure editing quality:

- **$\text{CLIP}_{\text{img}}$** measures the perceptual similarity between the base image and the edited image in the CLIP image-embedding space. This captures overall content preservation, ensuring that changes do not unintentionally alter unrelated aspects of the scene.

- **$\text{CLIP}_{\text{txt}}$** measures the alignment between the edited image and the concept prompt in the CLIP text–image embedding space. This captures the global semantic strength of the inserted concept.

- **$\text{CLIP}_{\text{dir}}$** compares the image-edit direction (base $\rightarrow$ edited image) with the textual direction (base prompt $\rightarrow$ concept prompt). Rather than isolating the concept, this metric evaluates whether the semantic shift produced by the edit is consistent with the semantic shift implied by the prompts. High **$\text{CLIP}_{\text{dir}}$** indicates that the deviation introduced by editing is aligned with the intended conceptual change, rather than being arbitrary drift.

Together, these three metrics provide a more complete quantification of the insertion–preservation trade-off, and the results are consistent with the temporal patterns reported in the main analysis. Specifically, across increasing CIS intervention bands, we observe a systematic decrease in **$\text{CLIP}_{\text{img}}$**, indicating reduced content preservation as the edit is applied earlier in the denoising process. At the same time, both **$\text{CLIP}_{\text{txt}}$** and **$\text{CLIP}_{\text{dir}}$** increase with higher CIS bands, reflecting stronger alignment with the target concept and a more consistent semantic shift relative to the intended prompt change. This temporal trend mirrors the insertion dynamics observed in our CIS curves: early interventions yield stronger concept insertion but at the cost of greater deviation from the base image. Importantly,

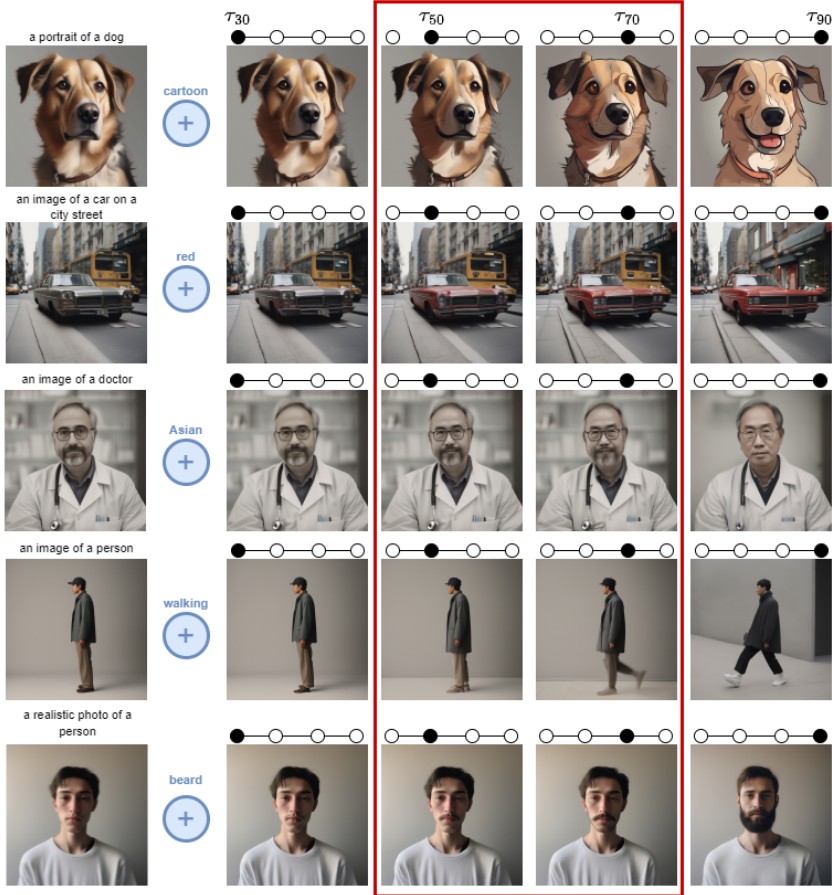

Figure 5: **Examples of text-driven image editing on SDXL.** The edited images are shown at four different points with their respective CIS probabilities: $\tau_{30}$, $\tau_{50}$, $\tau_{70}$, and $\tau_{90}$. High probabilities until a certain point ensure the intended modification but reduce preservation of the original image. We observe that CIS probabilities above 0.7 start to noticeably compromise the original content, and probabilities between 0.5 to 0.7 as suggested by our analysis (red rectangle) are best for editing while preserving the original image.

the CIS-guided window $[0.5, 0.7]$ consistently achieves the best balance between successful editing and fidelity across all metrics, and is therefore used as our main comparison setting. Tab. 1 shows that our method outperforms both NTI+P2P, as well as the recent state-of-the-art Stable Flow across all metrics. We report qualitative comparisons of text-driven image editing with respect to NTI+P2P and Stable Flow in Appendix Sec. D.5 (Fig. D7).

## 6 CONCLUSIONS

We introduced **Prompt-Conditioned Intervention (PCI)** as a simple, model-agnostic framework for probing *when* concepts can be injected during denoising under a fixed context. To complement this framework, we propose **Concept Insertion Success (CIS)** as the accompanying score that quantifies whether a concept intervention at a given timestep persists to the final image. For the interested reader we provide a short discussion of limitations in Appendix Sec. E. In summary, PCI and CIS turn diffusion time into an interpretable axis for analysis and editing: they expose context dependence, reveal cross-model stabilization differences, and provide actionable guidance on *when* to intervene. Our method and results yield a practical basis for temporally aware evaluation and more reliable, timing-sensitive editing of generative models.

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

# Temporal Concept Dynamics in Diffusion Models via Prompt-Conditioned Interventions

## Appendix

This supplement provides the complete set of technical details, implementation components, and extended analyses supporting the main paper. Appendix Sec. A covers methodological foundations of PCI, including diffusion model preliminaries, seed resampling, VQA templates, and the full formulation of CDS. Appendix Sec. B summarizes the experimental setup and evaluation protocol. Appendix Sec. C presents ablation studies assessing robustness across VQA models, seed counts, and prompt variations. Appendix Sec. D reports extended analyses such as fine-grained concept behavior, cross-attention visualizations, multi-concept interactions, abstract concepts, and additional qualitative editing results. Appendix Sec. E outlines the limitations of our approach.

## A  Implementation and Methodological Details

Here, we provide supplemental details on our PCI framework, including diffusion-model preliminaries (Appendix Sec. A.1), our concept-controlled seed resampling procedure (Appendix Sec. A.2), VQA question templates (Appendix Sec. A.3), and the full definition of Concept Deletion Success (CDS), the complementary measure to CI (Appendix Sec. A.4).

### A.1  Diffusion Model Preliminaries

Diffusion models generate images by *iteratively* refining noisy samples through a learned reverse process (Chen et al., 2025; Ho et al., 2020). The forward (diffusion) process is a fixed Markov chain that progressively adds Gaussian noise $\epsilon \sim \mathcal{N}(\mathbf{0}, \mathbf{I})$ to a clean latent input $\mathbf{x}_0$, producing a sequence $\{\mathbf{x}_t\}_{t=1}^T$. At each step, the Gaussian variance schedule $\{\beta_t\}_{t=1}^T$ determines the noise level, where

$$\alpha_t = 1 - \beta_t, \qquad \bar{\alpha}_t = \prod_{i=1}^t \alpha_i . \tag{A.1}$$

Thus, $\bar{\alpha}_t$ is the cumulative product of noise-retention factors that controls the signal-to-noise ratio (SNR) across timesteps. The forward process admits the closed form

$$\mathbf{x}_t = \sqrt{\bar{\alpha}_t}\, \mathbf{x}_0 + \sqrt{1 - \bar{\alpha}_t}\, \epsilon, \qquad t \in \{1, ..., T\}. \tag{A.2}$$

The generative model learns to reverse this chain by predicting the noise component $\epsilon$ from a given $\mathbf{x}_t$, using a neural network $\epsilon_\theta(\mathbf{x}_t, t, \mathbf{c})$ parametrized by $\theta$ and optionally conditioned on auxiliary input $\mathbf{c}$ (e.g., a text embedding). A reconstruction of the original data, denoted $\mathbf{x}_0^*$, can then be obtained by inverting the forward process:

$$\mathbf{x}_0^* = \frac{1}{\sqrt{\bar{\alpha}_t}} \left( \mathbf{x}_t - \sqrt{1 - \bar{\alpha}_t}\, \epsilon_\theta(\mathbf{x}_t, t, \mathbf{c}) \right) . \tag{A.3}$$

At inference time, the process begins by sampling a pure random Gaussian noise $\mathbf{x}_T$ and proceeds by iteratively denoising over timesteps until $\mathbf{x}_0$, which forms a synthesized image aligned with the conditioning input. For text-to-image generation, conditioning is injected via cross-attention layers, enabling semantic alignment between prompts and generated images.

To strengthen the alignment between conditioning inputs and generated content, diffusion models often employ classifier-free guidance (CFG) (Ho & Salimans, 2021). In this approach, an unconditional prediction $\epsilon_\theta(\mathbf{x}_t, t, \varnothing)$ is combined with the conditional prediction $\epsilon_\theta(\mathbf{x}_t, t, \mathbf{c})$ through a guidance scale $\omega$, yielding

$$\hat{\epsilon}_\theta = (1 + \omega)\, \epsilon_\theta(\mathbf{x}_t, t, \mathbf{c}) - \omega\, \epsilon_\theta(\mathbf{x}_t, t, \varnothing), \tag{A.4}$$

This extrapolation biases the denoising process toward the conditioning signal to further improve semantic fidelity.

### A.2  Concept-Controlled Seed Resampling

A key requirement of PCI is that generations obtained from the base prompt $P_b$ remain neutral with respect to the target concept, ensuring that switching to $P_c$ isolates the concept's influence on the trajectory. In practice, however, diffusion models may violate this requirement, sometimes rendering the concept even under $P_b$ (e.g., glasses appearing in a "neutral" face prompt). In addition to such prompt-level violations, models also exhibit structural priors that cause certain concepts to be overrepresented, regardless of whether they are specified in the prompt (e.g., mountain appearing in a "landscape" scene; see Fig. A1). Both effects undermine the contrast between $P_b$ and $P_c$ and lead to unreliable CIS functions. To mitigate



Figure A1: Per-concept VQA outcome fractions under the base prompt $P_b =$ "a realistic photo of a landscape", using SD 2.1 over 1000 seeds.

this issue, we propose a two-stage strategy prior to our analysis. First, we apply seed filtering to discard cases where the target concept is already present under $P_b$ using the same VQA model. If this fails to yield sufficient number of valid seeds, we incorporate negative guidance (Ho & Salimans, 2021) as an additional constraint, where the concept itself is used as a negative prompt to discourage its appearance during generation with $P_b$. This approach goes beyond naïve prompt engineering with terms such as "no" or "not," which have been shown to be ineffective (Alhamoud et al., 2025). Instead, it follows the CFG formulation, but with the guidance term inverted to steer the model away from the specified concept at each denoising step. While negative guidance has been shown to reduce undesired concepts, recent work such as Normalized Attention Guidance (NAG) (Chen et al., 2025) demonstrates that guidance alone is not sufficient to ensure stable outcomes. In our setting, we similarly find that negative guidance by itself does not provide reliable neutrality, but when combined with seed filtering it yields the most stable configuration. We therefore adopt this combined strategy in our main experiments.

## A.3 VQA QUESTION TEMPLATES

We present the prompts we used for our VQA model for the different subcategories in Table A1. For each subcategory, the same VQA prompt applies to all fine-grained concepts.

| Category | Subcategory | Question |
|---|---|---|
| Demographics | Gender | Is the person in the image a <concept>? |
| | Ethnicity | Is the person in the image <concept>? |
| | Age group | Is the person in the image a <concept>? |
| Objects | Animals | Is there a <concept> in the image? |
| | Handmade objects | Is there a <concept> in the image? |
| | Nature objects | Is there a <concept> in the image? |
| Human properties | Clothing | Is the person in the image wearing a <concept>? |
| | Accessories | Is the person in the image wearing <concept>? |
| | Physical traits | Does the person in the image have <concept>? |
| Actions | Common actions | Is the person in the image <concept>? |
| Attributes | Material | Is the table in the image made of <concept>? |
| | Color | Is the table in the image <concept>? |
| Environment | Season | Is the season <concept>? |
| | Weather | Is the weather <concept>? |
| | Time | Is it <concept>? |
| | Setting / Venue | Is the photo taken in a <concept>? |
| Style | Image style | Is the style of the image <concept>? |

Table A1: Overview of dataset categories, subcategories, and associated questions.

## A.4 CONCEPT DELETION SUCCESS (CDS)

While CIS measures the earliest timestep at which a concept can be *inserted* into the denoising trajectory, an equally important question is: *when does a concept become unavoidable?* In other words, at what point in the trajectory is a concept already committed to the generation, such that later attempts to remove it no longer succeed? We formalize this complementary perspective through the **Concept Deletion Success (CDS) score**. Given a concept prompt $P_c$ and a base prompt $P_b$ that omits the target concept, CDS quantifies the probability that switching the conditioning *from $P_c$ to $P_b$* at timestep $t_s$ successfully removes the concept from the generated sample. Intuitively, CDS characterizes the *latest* timestep at which overriding the concept remains effective. Beyond this point, the concept becomes *locked in*: even reverting to $P_b$ cannot eliminate it from the generation. Formally, for each timestep $t_s$, we construct the reconstruction

$$\mathbf{x}_0(P_c \xrightarrow{t_s} P_b), \tag{A.5}$$

which follows the concept prompt $P_c$ until timestep $t_s$, and subsequently denoises under the base prompt $P_b$. To evaluate whether the target concept $c$ persists in the resulting image, we again employ

Qwen-3B in a VQA setup. For each reconstruction, the VQA model outputs a binary decision indicating whether the concept is present ("yes") or absent ("no"). We define the **Concept Deletion Score** for all timesteps computed across random seeds. Low CDS values indicate successful concept removal, whereas high CDS values indicate that the concept persists despite the attempted deletion. By evaluating CDS over $t_s = T \rightarrow 0$, we can identify the *deletion locking point*, i.e., the latest timestep at which a concept becomes resistant to removal. Together with CIS, CDS provides a comprehensive characterization of concept formation, stability, and irreversibility during the denoising process.

Empirically, we observe that deletion dynamics behave *differently* from insertion dynamics, but not in a universally uniform manner. Figs. A2 and A3 show the Concept Deletion Success (CDS) trajectories for SD 2.1 and SDXL for a representative concept–base-prompt pair, averaged over 100 random seeds. While CIS curves exhibit substantial variability in when different concepts become insertable, CDS curves reveal a distinct trend: concept deletion tends to fail considerably earlier in the denoising trajectory. However, there is still noticeable variation across concepts, categories, and prompt configurations, indicating that deletion is not governed by a single universal threshold. Two factors help explain this behaviour. First, when generation begins under the concept prompt $P_c$, the model is effectively primed toward that concept from the earliest steps, making subsequent removal more difficult and often shifting the effective deletion window to earlier timesteps. Second, certain concept pairs (e.g., binary attribute switches such as *night vs. morning*) behave differently from cases where a concept is simply present or absent, leading to sharper or more gradual transition behaviors depending on semantic structure.

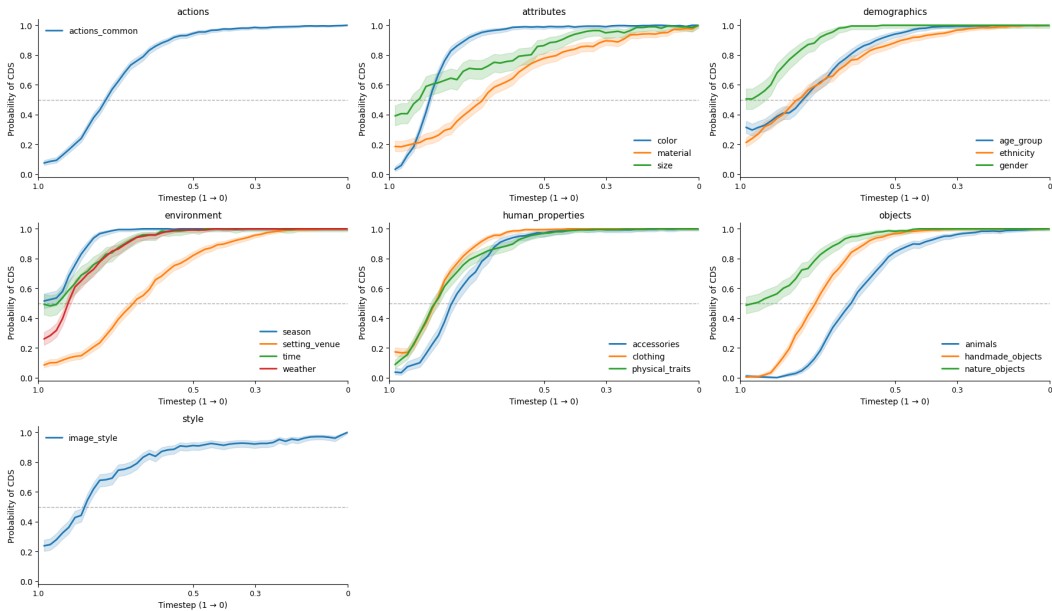

Figure A2: Concept Deletion Success (CDS) curves on SD21 for all concepts for a given concept-base prompt pair.

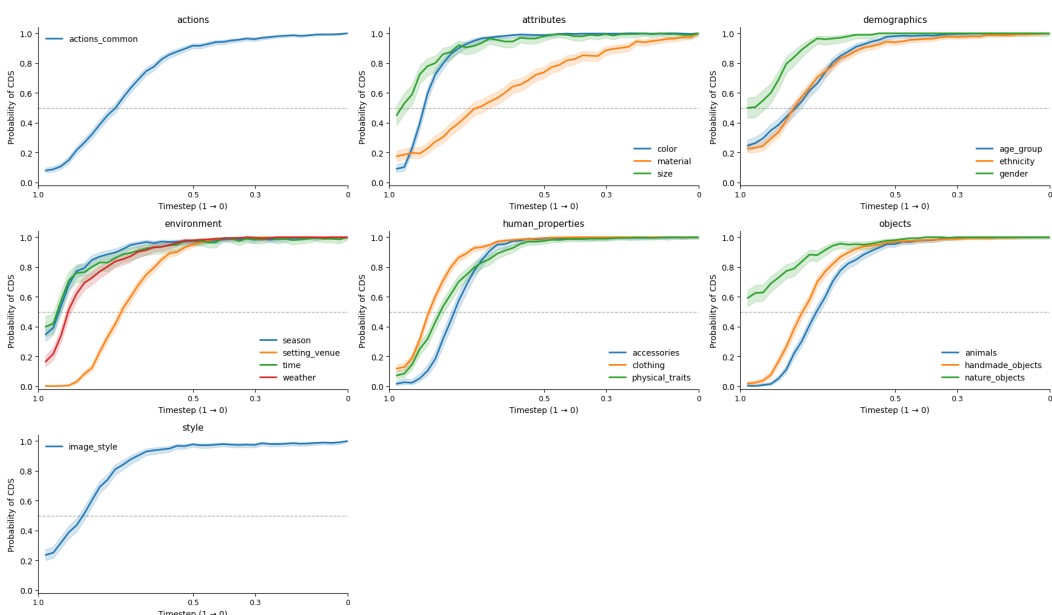

Figure A3: Concept Deletion Success (CDS) curves on SDXL for all concepts for a given concept-base prompt pair.

## B    EXPERIMENTAL SETUP AND EVALUATION PROTOCOL

Here, we summarize the experimental settings used throughout our study. We describe the models and inference configurations, including cross-model timestep normalization (Appendix Sec. B.1), and outline the evaluation metrics applied to quantify temporal concept behavior (Appendix Sec. B.2).

### B.1    MODEL DETAILS AND CROSS-MODEL TIMESTEP NORMALIZATION

For diffusion-based models, we evaluate Stable Diffusion 2.1 (SD 2.1) Rombach et al. (2022a) and Stable Diffusion XL (SDXL) Podell et al. (2024). For flow-matching models, we evaluated Stable Diffusion 3.5 (SD 3.5) Esser et al. (2024b); Sauer et al. (2024). Unless otherwise mentioned, we use Qwen-VL-3B Bai et al. (2025) as the VQA scorer for CIS and report a model ablation in Appendix Sec. C.1 with other VQA models. We keep the original scheduler and CFG settings released for each Stable Diffusion model we used without modification. For all our experiments, we use 100 seeds. We refer to Appendix Sec. C.2 for an ablation of the number of seeds required to establish a robust CIS. All models traverse the diffusion trajectory between timestep range $t \in [0, 1000]$ (early $t=1000$ to late $t=0$) with model-specific step counts $T$. SD 2.1 and SD 3.5 use $T=50$ inference steps, whereas SDXL employs $T=40$ steps. For cross-model comparability, we normalize all timesteps to the range [0,1].

All models traverse the diffusion trajectory between timestep range $t \in [0, 1000]$ (early $t=1000$ to late $t=0$) with model-specific step counts $T$:

$$t_k = 1000 - k\,\Delta t, \quad k = 0, \ldots, T, \qquad \Delta t = \tfrac{1000}{T}, \tag{B.1}$$

Concretely, SD 2.1 and SD 3.5 use $T=50$ inference steps, corresponding to a step size of $\Delta t = 20$, whereas SDXL employs $T=40$ steps with $\Delta t = 25$. For the rest of cross-model comparability we refer the timestep axis as $\tau := t/1000 \in [0, 1]$.

### B.2    EVALUATION METRICS

Let $C(\tau) \in [0, 1]$ be the CIS curve. For $q \in \{0.50, 0.70\}$ we define the crossing time as $\tau_q = \min\{\tau \in [0, 1] : C(\tau) \geq q\}$, which is the smallest $\tau$ that achieves CIS probability $q$. We also measure how quickly a concept locks within the analysis band via the *bandwidth-steepness*: $W_{70 \to 50} = \tau_{70} - \tau_{50}$, where smaller $W$ indicates a sharper, less timing-fragile transition inside the 0.50–0.70 band. We chose $q \in \{0.50, 0.70\}$ because a CIS of 0.50 marks the indifference (coin-flip) point where a concept becomes as likely as not to insert successfully, and 0.70 serves as a high-confidence operating point for reliable insertions without requiring near-certainty.

**Statistical aggregation and uncertainty.** We evaluate CIS for each concept at each timestep under a fixed base prompt (the *context*) into which the concept is introduced. For every concept–prompt pair, we run multiple random seeds and average the resulting binary VQA outcomes across seeds to obtain a smooth CIS curve over timesteps. This removes seed-level noise and yields one representative curve per concept–prompt. To summarize *when* insertion starts to succeed, we analyze the timesteps at which the averaged curve first reaches CIS target levels (e.g., 50% and 70%); if a level is never reached, the corresponding time is left undefined. For cross-model comparisons within a subcategory $\mathcal{S}$ (e.g., all prompts and concepts in "animals" or "weather"), we report the mean of these crossing times across all concept–prompt pairs in $\mathcal{S}$, together with their standard deviation across pairs.

For visualization, we also aggregate the seed-averaged CIS curves across all pairs in $\mathcal{S}$ to show an overall trend, and overlay the individual curves to display variability across prompts/contexts. When focusing on a single concept–prompt pair, uncertainty primarily comes from finite seeds; we therefore display 95% Wilson confidence intervals around the seed-averaged CIS at each timestep. In short, Wilson intervals quantify *seed-level* uncertainty for a given pair, while the spread of curves (and the reported mean ± std) captures *prompt/context-level* variability across the subcategory.

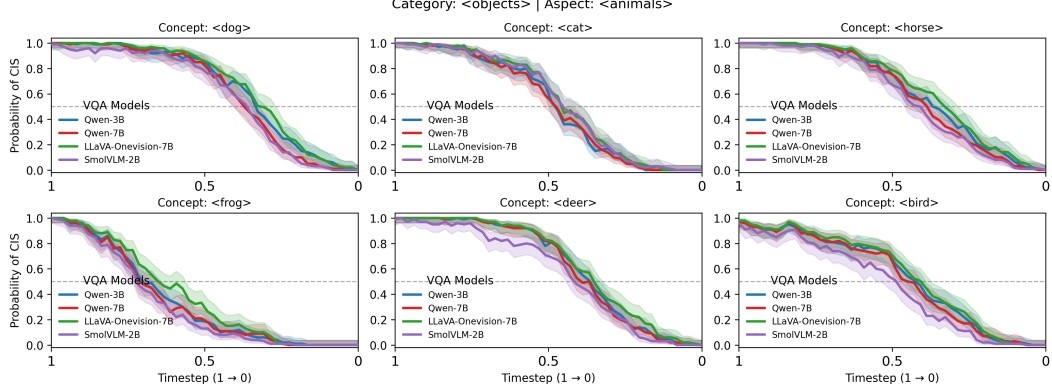

Figure C1: **Ablation studies across different LVLM models**. We show the VQA evaluation of the CIS results for 6 different concepts from animals subcategory, showing that CIS results remain consistent across different LVLM models: Qwen-3B, Qwen-7B, LLaVA-OneVision-7B, and SmolVLM-2B.

## C  ABLATION STUDIES

Here, we present ablation experiments that assess the robustness of our framework. We evaluate the consistency of CIS across different VQA scorers (Appendix Sec. C.1), analyze sensitivity to the number of random seeds (Appendix Sec. C.2), and test stability under natural prompt rephrasings (Appendix Sec. C.3).

### C.1  CROSS-MODEL VQA CONSISTENCY ANALYSIS

To ensure our results are not an artifact of a single LVLM, we have expanded our VQA evaluation to include four models: Qwen-VL-2.5 3B, Qwen-VL-2.5 7B, LLaVA-OneVision-7B Li et al. (2025), and SmolVLM-2B Marafioti et al. (2025). Specifically, we conducted experiments on the animals category using SD-2.1 with identical prompts in Fig. C1, in which the responses of all models are reported. Across all concepts, the models produce closely matching CIS trajectories, demonstrating that PCI is agnostic and robust to the specific LVLM model in our setting, *and that even small models (SmolVLM-2B) achieve comparable results*. Given the comparable outcomes and the lower computational cost, we adopt Qwen-VL 3B for the main experiments.

### C.2  SAMPLE SIZE SENSITIVITY ANALYSIS

We evaluated how many seeds are required to estimate reliable CIS curves in a controlled experiment using SD-2.1.

For the selected concepts, we first run PCI to obtain CIS trajectories for 1000 distinct seeds using SD 2.1. To quantify how many seeds are needed, we repeatedly subsample $k \in \{1, \ldots, 1000\}$ seeds (100-bootstrap resampling) and compute the mean CIS trajectory and its variance across samples. Stability is then assessed by requiring (i) low bootstrap variance and (ii) consistency with the mean curves obtained at larger $k$. For each concept, we record the smallest $k$ that satisfies the stability criteria and aggregate these between concepts to obtain a coverage–vs–$k$ curve (fraction of concepts stable at each $k$). Based on this curve, we set $k = 100$ for all experiments, as increasing the seed budget beyond does not appreciably raise coverage.

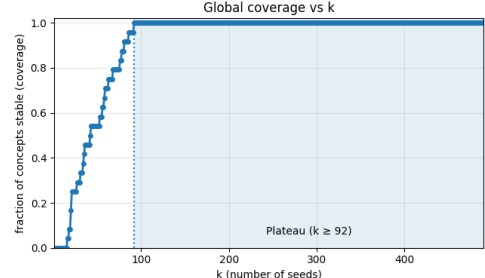

Figure C2: Seed budget analysis for estimating stable CIS trajectories.

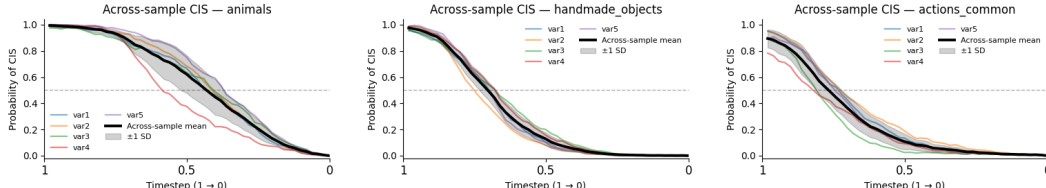

Figure C3: **Prompt robustness across semantic variations.** We evaluate the stability of CIS trajectories under prompt variations for three representative subcategories: animals, handmade objects, and common actions. For each subcategory, we generate five semantically equivalent prompt variants and compute CIS curves averaged over all concepts and seeds. The resulting trajectories illustrate the robustness of our method to natural prompt rephrasings: while absolute CIS values may shift slightly across variants, the overall temporal patterns and locking behaviours remain consistent, demonstrating that our findings are not tied to a specific wording of the base or concept prompts.

## C.3 BASE PROMPT ROBUSTNESS ANALYSIS

A potential failure mode of prompt-based interpretability analyses is that they may capture prompt phrasing effects rather than genuine properties of the diffusion trajectory. To ensure that CIS reflects the intrinsic temporal behavior of the target concept and not artifacts of a particular linguistic formulation, we perform a **prompt robustness evaluation** using SD 2.1. Given a base–concept prompt pair $(P_b, P_c)$ for a target concept $c$, we construct a small family of paraphrased variants obtained using GPT-5 (see Table C1), where each variant preserves the semantic context of the original pair while introducing minor syntactic or descriptive changes (e.g., "an image of a person", "a photo of a person"). These paraphrases are designed such that they neither introduce new visual attributes nor alter the underlying meaning of the target concept. For each variant, we recompute the CIS curve and compare those results with the original CIS curve.

As illustrated in Fig. C3, the resulting CIS trajectories are highly consistent across the five paraphrased variants for representative subcategories such as *animals*, *handmade objects*, and *common actions*. While slight variations in absolute CIS values can occur due to wording-specific stylistic biases, the overall temporal profiles, including the onset, rise, and stabilization regions, remain stable. Crucially, all variants exhibit the same relative ordering of insertion windows across timesteps, indicating that the observed temporal dynamics stem from the model's generative behavior rather than from linguistic artifacts. These results confirm that CIS is robust to natural prompt rephrasings and reliably captures concept-specific temporal behavior independent of the exact phrasing of $(P_b, P_c)$.

| Subcategory | Base Prompts |
|---|---|
| Animals | 1) **an image of a park** 
 2) a photo of a park 
 3) a visual depicting a park 
 4) a detailed view of a park 
 5) a realistic photo of a park |
| Handmade Objects | 1) **an image of an empty scene** 
 2) a photograph of an empty scene 
 3) a visual depicting an empty scene 
 4) a detailed view of an empty scene 
 5) a realistic photo of an empty scene |
| Common actions | 1) **an image of a person** 
 2) a picture of a person 
 3) an illustration of a person 
 4) a high-resolution image depicting a human figure without any explicit activity 
 5) a depiction of a person in a neutral setting |

Table C1: **Overview of the prompt variations.** For the selected subcategories, we construct a total of five prompts: the original base prompt (shown in **bold**) and four semantically consistent paraphrased variants used to assess prompt robustness.

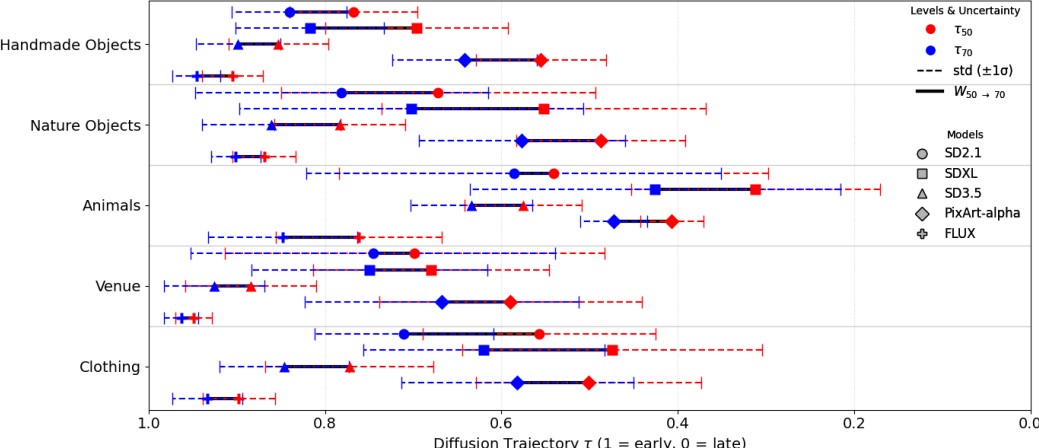

Figure D1: **CIS reveals cross-concept and cross-architecture differences.** CIS for ● $\tau_{50}$ and ● $\tau_{70}$ across multiple concept categories and diffusion models.

## D  EXTENDED ANALYSES AND ADDITIONAL RESULTS

Here, we report extended results complementing the main analysis, including additional global analyses results for the rest of the categories in Fig. D1.. We provide fine-grained concept and context studies (Appendix Sec. D.1), cross-attention visualizations (Appendix Sec. D.2), multi-concept interaction analyses (Appendix Sec. D.3), results on abstract concepts (Appendix Sec. D.4), and additional qualitative examples of text-driven image editing (Appendix Sec. D.5).

### D.1  FINE-GRAINED CONCEPT ANALYSIS

While category-level summaries highlight broad timing patterns, they can mask the specific prompts (contexts) and concepts that underlie these patterns. Fine-grained analyses investigates these sources of variation by showing full CIS curves at the level of (i) a fixed concept across different prompts (prompt-level), and (ii) different concepts within the same subcategory under matched prompts (concept-level). This reveals when the aggregated behavior of a category is dominated by a subset of contexts or a subset of its concepts and highlights *meaningful concept–context relationships* that are obscured by averaging (e.g., how the same concept behaves across distinct settings, or how different concepts within a category respond to the same context setting).

Here, we carry out this fine-grained analysis for SDXL. For a *prompt-level* comparison, we fix a subcategory and vary the base prompt (context) for plotting CIS curve with **95% Wilson CIs** over seeds. For a *concept-level* comparison, we fix the prompt and plot the CIS curve for several concepts within the same subcategory with **95% Wilson CIs** over seeds.

> **Finding: Scene–concept alignment governs timing:** When the concept aligns with the context, both $\tau_q$ and $W$ are smaller (later and more robust specification). Out-of-distribution pairings inflate $\tau_q$ (early-only window) and often $W$ (greater timing fragility), unless model capacity compresses $W$.

**Prompt-level analysis (Fig. D2a).**  In Fig. D2a, we demonstrate representative prompt-level effects of the derived CIS function for "style", "animals", "age group" and "ethnicity" categories, under different two different contexts. Starting with a broad category of style, Fig. D2a, top-left, shows the CIS plots for the contexts kitchen interior (blue curve) and bowl of fruit (red curve). Under the context of bowl of fruit, the influence of style emerges earlier in the diffusion trajectory but rapidly diminishes to zero $\tau \approx 0.5$, whereas for the kitchen interior, style persists longer in the trajectory. For animals, Fig. D2a, bottom-left reveals that under the context of outdoor environment (e.g., countryside), animals can be incorporated into the trajectory much later than those under the context of indoor environment (e.g., living room) as it is generally harder to put wild animals within an indoor OOD context. In contrast, some categories, such as age, exhibit close dynamics across different contexts, resulting in closely aligned CIS plots as shown in Fig. D2a, bottom-right.

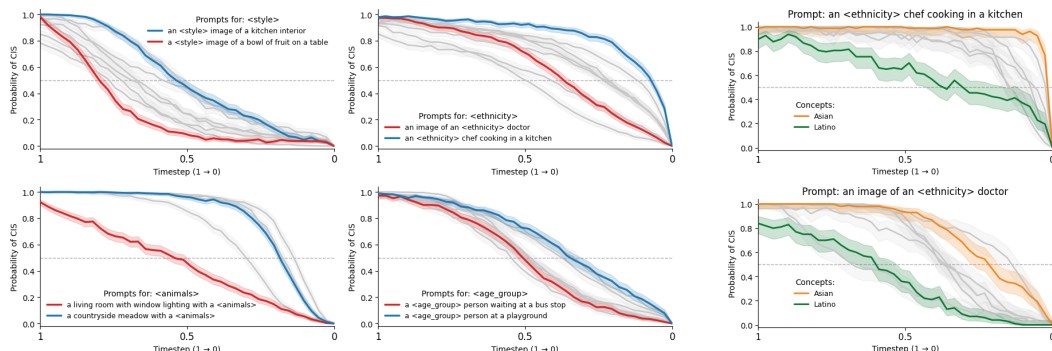

(a) Category-level CIS dynamics for: *style, ethnicity, animals, age group*.

(b) Concept-level CIS dynamics for *ethnicity*.

Figure D2: **CIS dynamics.** (a) For each category, we plot the probability of Concept Insertion Success (CIS) across the diffusion trajectory ($1 \rightarrow 0$, noisy $\rightarrow$ clean) for multiple *base prompts*, averaged over all concepts in that category. Two representative prompts are highlighted (blue and red), showing that the context in the base prompt can affect the concept insertion. (b) For ethnicity, we decompose the CIS functions of the two prompts into their individual elements (*fine-grained concepts*) variations, with two different concepts highlighted (green and orange), demonstrating that impact of individual concepts on the category-level dynamics. Overall, CIS functions allow us to understand the temporal evolution of broad categories and fine-grained concepts in diffusion models.

**Concept-level analysis (Fig. D2b).** In Fig. D2b, we further decompose the CIS functions for the category "ethnicity" under different contexts, into its different elements (the fine-grained concepts of the category "ethnicity"). We show two such concepts: "Asian" (orange curve) and "Latino" (green curve), under two different contexts: "chef in a kitchen" and "doctor". This allows us to understand the impact of individual concepts on the "ethnicity" category. Certain ethnicities such as "Asian" appear very late in the diffusion trajectory compared to others, and are also affected by different contexts. Overall, the CIS function allows us to understand temporal evolution of category-level and concept-level dynamics in diffusion models.

## D.2    CROSS-ATTENTION VISUALIZATION STUDIES

To complement the binary Concept Insertion Success (CIS) metric, we additionally perform a *qualitative* analysis based on cross-attention maps of SDXL. While CIS indicates whether a concept appears in the final image, it does not provide information about concept strength or spatial manifestation. The cross-attention analysis offers a complementary, spatially grounded perspective.

Given a specific $\tau$ value, we follow Tang et al. (2022) to extract cross-attention maps. Specifically, for a given concept word $c$, we extract token-level cross-attention maps from all U-Net cross-attention layers during image generation. We perform this for all timesteps, including the timesteps before and after the intervention. We then average the cross-attention maps over all timesteps. This leads to a single heatmap representing intervention at $\tau$. We repeat this setup for several $\tau$ values: $\tau_0$, $\tau_{30}$, $\tau_{50}$, $\tau_{60}$, $\tau_{70}$ and $\tau_{90}$. We then take the global minimum and maximum across all the resulting $\tau$ cross-attention maps, and normalize each to the range of 0-1, using those minimum and maximum statistics.

We apply PCI at several representative CIS probabilities ($\tau_0$, $\tau_{30}$, $\tau_{50}$, $\tau_{70}$, $\tau_{90}$) and visualize how the attention for token $c$ evolves as the intervention timestep changes. This procedure enables inspection of (i) which spatial regions are associated with the concept, (ii) how concentrated or diffuse the attention becomes, and (iii) how these patterns vary across the denoising trajectory.

As illustrative examples, Fig. D3 shows how the spatial activation of the concept token evolves across different CIS bands for several representative concepts. Although we do not provide quantitative measurements, we observe consistent qualitative tendencies. Interventions within higher-CIS regions often lead to clearer and more spatially focused cross-attention associated with the concept, while interventions within lower-CIS regions show weaker or less coherent activations. These patterns offer

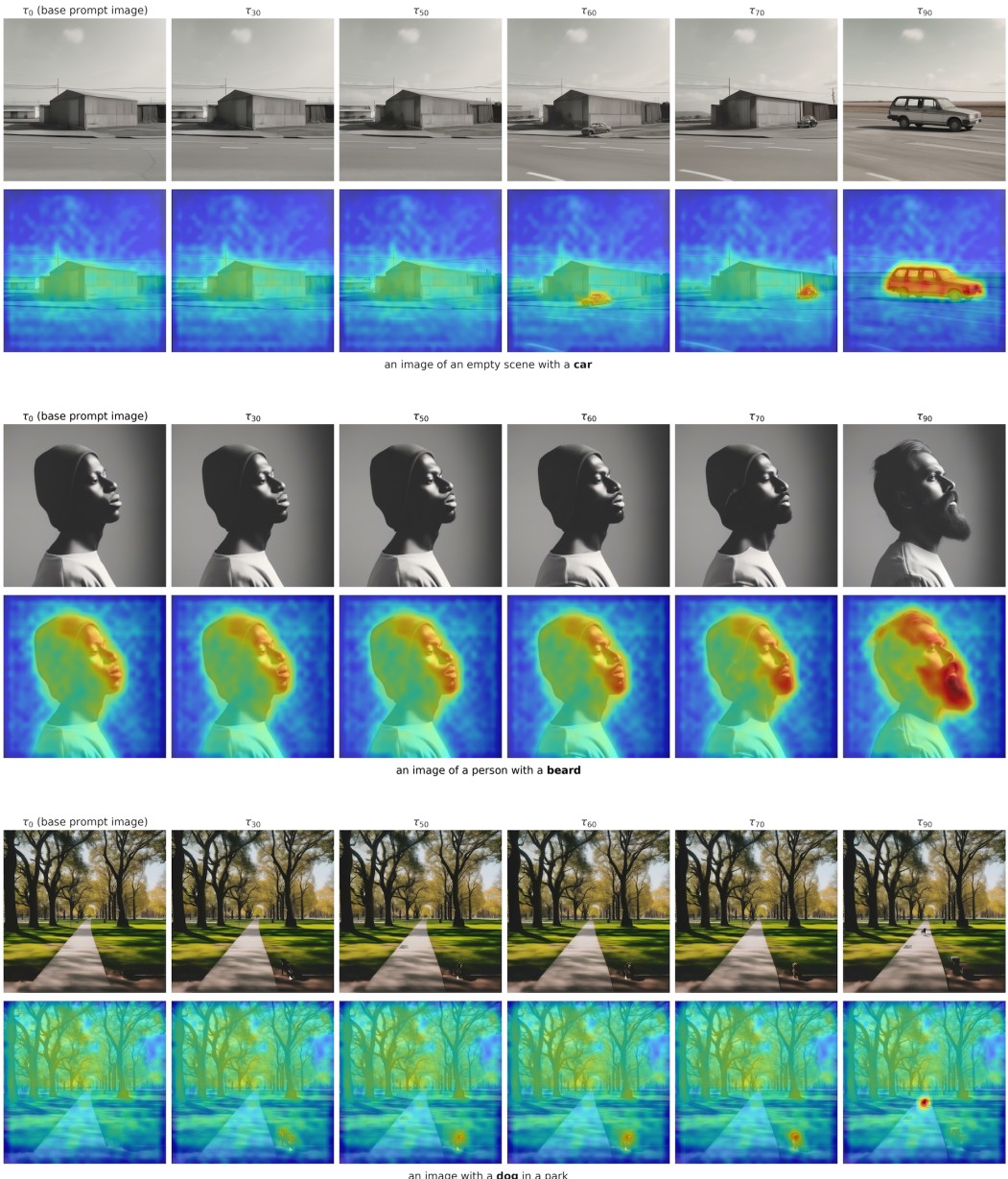

Figure D3: **Qualitative cross-attention maps for concept token** $c$ **at different intervention timesteps.** Attention becomes stronger and more spatially coherent in high-CIS regimes and weaker or diffuse in low-CIS regimes, providing a visual complement to the binary CIS metric.

intuitive insight into how concept strength and spatial coherence change as insertion becomes more or less feasible.

### D.3    MULTIPLE CONCEPT INTERACTION ANALYSIS

While our primary analysis focuses on single-concept insertion through PCI, many real-world editing scenarios require simultaneously modifying or introducing multiple attributes.To study these interactions, we extend PCI to the *multi-concept* setting using SD 2.1, where two concepts are introduced simultaneously at a target timestep $t_s$. For each concept pair $(c_1, c_2)$, we construct CIS curves by switching from the base prompt $P_b$ to the combined prompt $P_{c_1,c_2}$ at different timesteps.

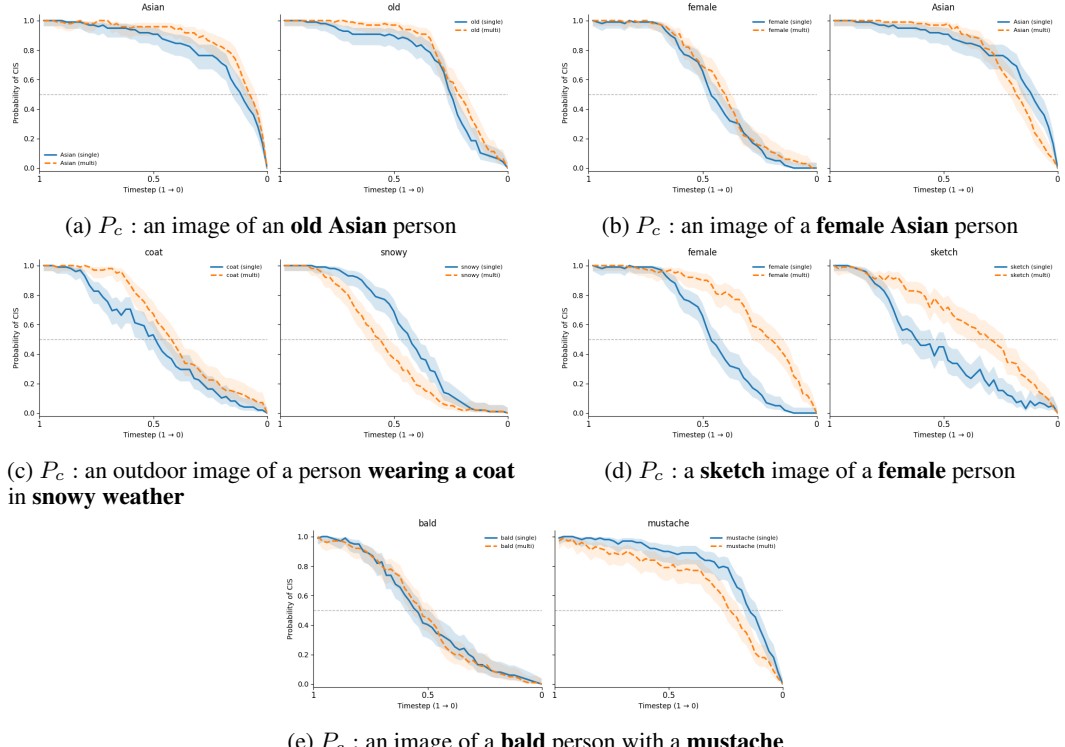

(a) $P_c$ : an image of an **old Asian** person

(b) $P_c$ : an image of a **female Asian** person

(c) $P_c$ : an outdoor image of a person **wearing a coat** in **snowy weather**

(d) $P_c$ : a **sketch** image of a **female** person

(e) $P_c$ : an image of a **bald** person with a **mustache**

Figure D4: **Comparison of CIS curves under single- and multi-concept conditioning across five samples.** Each subfigure contains two evaluations: (left) CIS of concept $c_1$ under its single-concept prompt vs. CIS of $c_1$ under the joint $(c_1 + c_2)$ prompt; (right) the corresponding evaluation for concept $c_2$. Although both subplots use the same multi-concept prompt, the target concept differs, resulting in different CIS trajectories. Most pairs exhibit near-compositional behavior (a,b,c,e), whereas others (d) show interaction-dependent shifts in insertion timing.

In this setting, CIS naturally remains a *concept-specific* measure: even if the conditioning prompt includes multiple concepts, insertion success is always evaluated with respect to one target concept at a time.

$$\text{CIS}(c_1 \mid P_b \to P_{c_1}) \quad \text{vs.} \quad \text{CIS}(c_1 \mid P_b \to P_{c_1, c_2}), \tag{D.1}$$

which capture how each concept behaves when introduced alone versus jointly. Although both evaluations use the same multi-concept prompt, the target concept differs, and therefore the resulting CIS trajectories may also differ.

This formulation enables us to identify asymmetries: one concept may become temporally fixed earlier than another, or one may remain editable while the other is already locked in. Empirically, many concept pairs behave in a near-compositional manner (e.g., *coat + snowy* and *female + Asian*) where the joint CIS curves closely follow their single-concept baselines. However, some combinations exhibit clear interaction effects. For instance, the pair *female + sketch* shows a substantial shift of the insertion window toward later timesteps, allowing both attributes to remain editable deeper in the denoising trajectory than either concept alone. This demonstrates that certain concept pairs can meaningfully influence one another's temporal insertion dynamics.

## D.4 ABSTRACT CONCEPTS ANALYSIS

Subjective or abstract concepts such as *elegant*, *art*, and *creative* can pose a challenge for LVLMs. These concepts are inherently subjective, and their interpretation may vary across cultures, contexts, or viewpoints. To better understand how the models behave in this setting, we computed CIS curves in Fig. D5 for these three concepts across all four LVLMs used in our study.

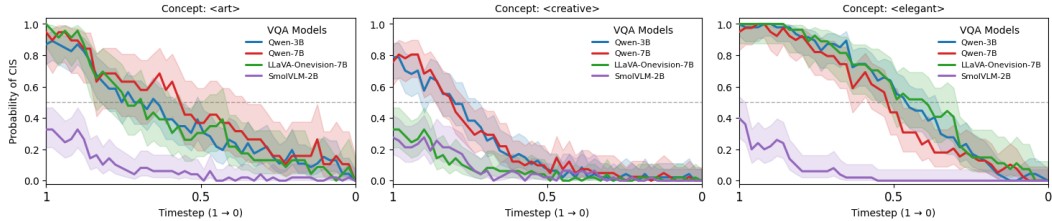

Figure D5: CIS curves for a single base–concept prompt pair involving the abstract concepts art, creative, and elegant.

Across the three abstract concepts, the three larger models (Qwen-3B, Qwen-7B, and LLaVA-OneVision-7B) display highly consistent behaviour: their CIS trajectories are smooth, monotonic, and closely aligned throughout the denoising process. In contrast, SmolVLM-2B consistently produces lower CIS values and a steeper decline, indicating a weaker ability to validate these subjective attributes when inserted later in the trajectory. Nevertheless, its CIS curves still follow the same qualitative pattern. In the raw CIS curves with Qwen-3B, concepts such as *creative* and *art* begin with higher initial CIS (around 0.8) but drop below 0.5 much earlier in the timestep sequence than *elegant*. This suggests that the LVLM has its own internal notion of what "elegant," "art," or "creative" means, shaped by the data it was trained on. The multi-model comparison confirms that these interpretations are broadly aligned across strong LVLMs, with only the smallest model deviating in magnitude but not in trend.

To account for subjectivity of individual models, we could perform multi-model agreement. Specifically, we can aggregate CIS curves from multiple VLMs (which have been trained on different data) to calibrate the CIS curves. We have added a section on multi-model agreement of four VLMs in Sec. C.1. If subjective ratings of an abstract concept are desired, we could instead extend the prompt to reflect the subjectiveness. For example, in the prompt "In Eastern Asia, would this image be considered elegant?", Eastern Asia provides the desired subjective context in which a user might want to evaluate CIS.

### D.5 QUALITATIVE RESULTS OF TEXT-DRIVEN IMAGE EDITING

We show comparison of editing performance of our method at $\tau_{60}$, compared to NTI-P2P Mokady et al. (2023) and Stable-Flow Avrahami et al. (2025) baselines in Fig. D7. Our method (PCI) produces edits that are semantically stronger, more spatially localized, and better aligned with the target concept while preserving non-edited regions. Moreover, in Fig. D6, we provide additional qualitative results for the application of text-driven image editing on SD 3.5.

## E LIMITATIONS

Every study has limitations, and ours is no exception. We highlight two main ones. **(1) Runtime:** Computing a CIS curve for a single seed depends on the number of interruption points at which we switch prompts. We used all native inference steps to achieve the highest resolution, but the process can be accelerated by sampling fewer points. Importantly, averaging across multiple seeds allows a sparsely sampled curve to approximate a densely sampled one. **(2) Editing sensitivity:** There is no single CIS value that reliably works for text-driven editing. Instead, a range, typically CIS probabilities between $0.5$ and $0.7$, performs best across categories and concepts. A practical workaround is to choose a general value that works reasonably well overall, even if not optimal for every case.

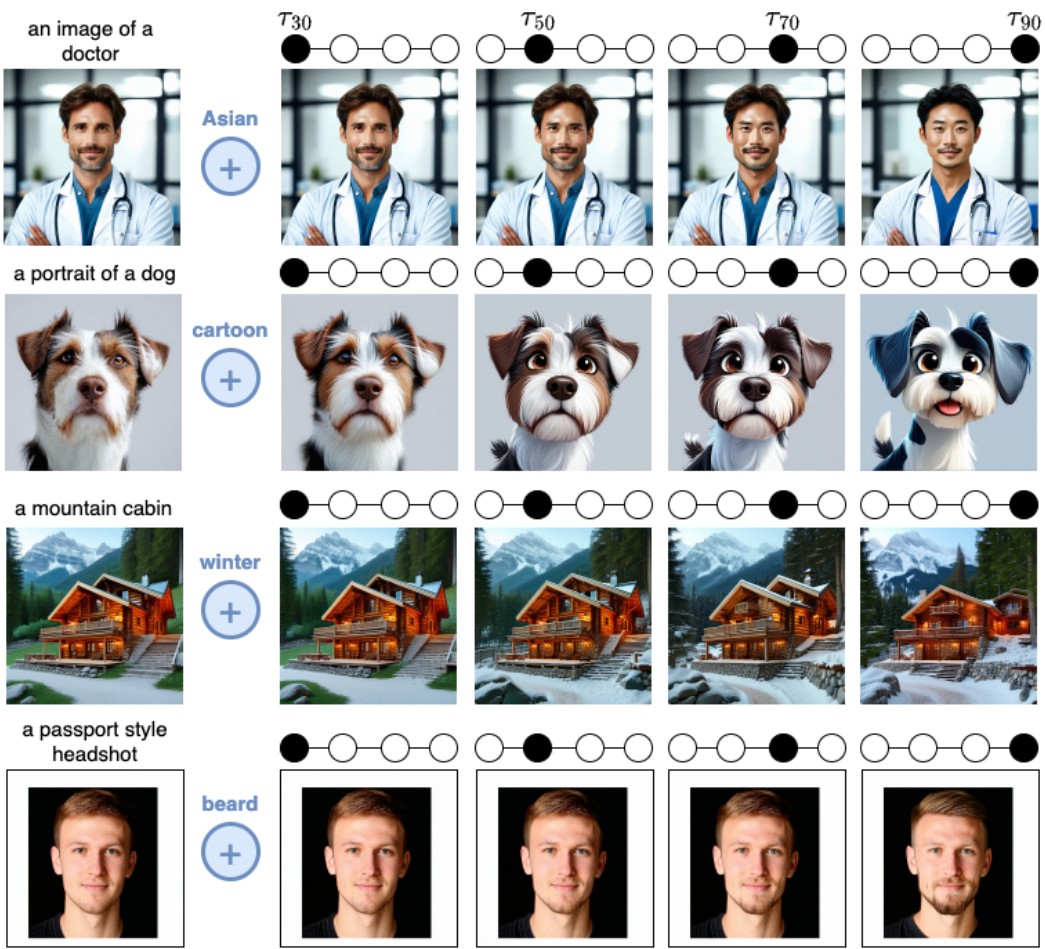

Figure D6: Examples of text-driven image editing on SD 3.5.

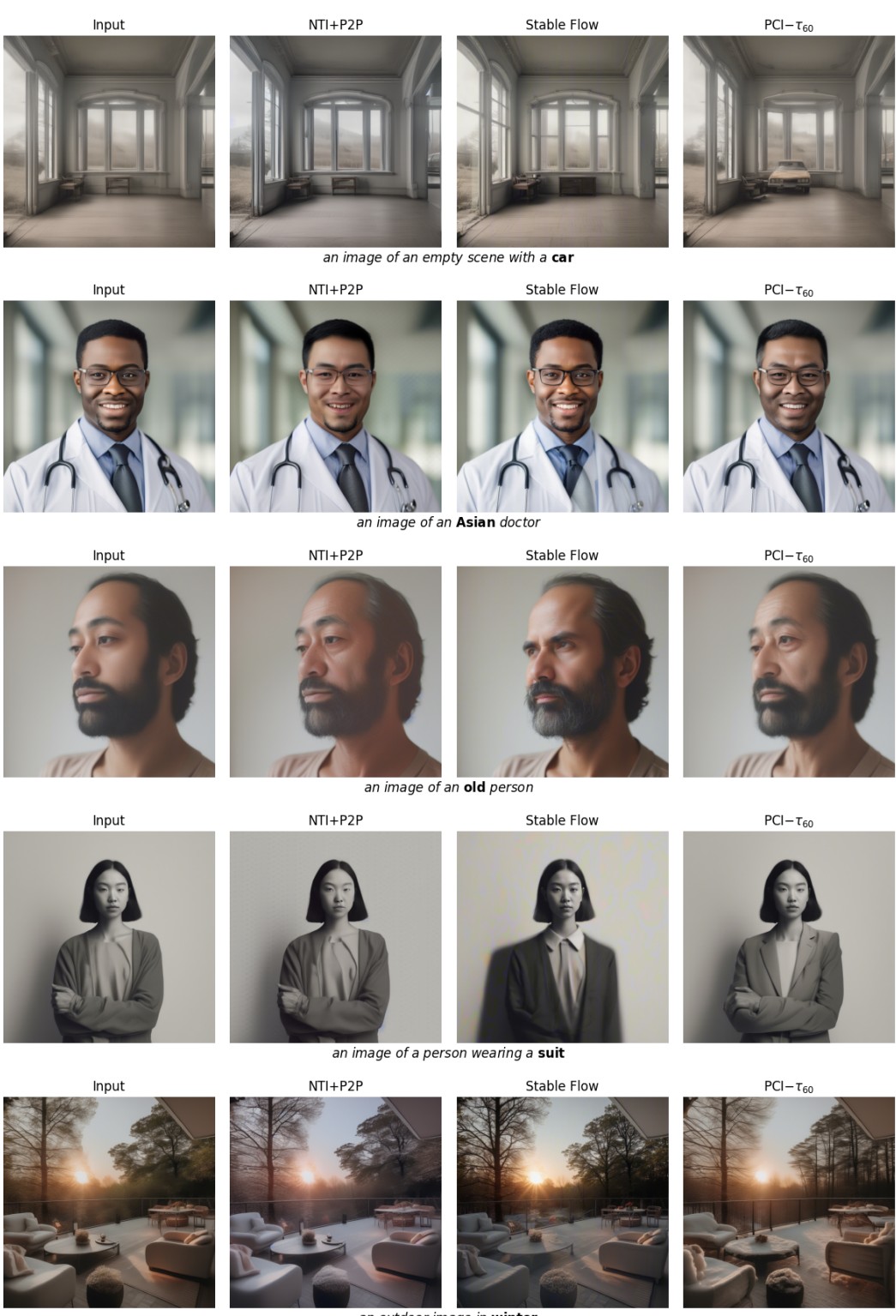

Figure D7: Comparison of editing performance across NTI-P2P, Stable-Flow, and our PCI method at $\tau_{60}$. PCI produces edits that are semantically stronger, more spatially localized, and better aligned with the target concept while preserving non-edited regions.

# Acknowledgment

Fawaz Sammani is funded by the Fonds Wetenschappelijk Onderzoek (FWO) (PhD) fellowship strategic basic research 1SH7W24N). N. Deligiannis acknowledges the "Onderzoeksprogramma Artificiele Intelligentie (AI) Vlaanderen" programme and the ERC Consolidator Grant IONIAN (No. 101171240, DOI: 10.3030/101171240). Funded by the European Union. Views and opinions expressed are however those of the author(s) only and do not necessarily reflect those of the European Union or the European Research Council Executive Agency. Neither the European Union nor the granting authority can be held responsible for them.

