# OpenReview forum: "Temporal Concept Dynamics in Diffusion Models via Prompt-Conditioned Interventions"
_ICLR.cc/2026/Conference — ICLR 2026 Poster_

### Official Review · Reviewer_uBmP · 2025-10-25

**Soundness:** 4
**Presentation:** 3
**Contribution:** 2
**Rating:** 4
**Confidence:** 5

**Summary:**

This paper proposes Prompt-Conditioned Intervention (PCI) to study when a semantic concept becomes established during the denoising process of diffusion and flow models. By switching prompts at different timesteps and using a vision-language model to check concept presence, the authors derive a Concept Insertion Success (CIS) curve that quantifies concept insertability over time. Experiments on Stable Diffusion 2.1, SDXL, and SD 3.5 show that global scene factors appear earlier, human attributes emerge mid-trajectory, and accessories lock in later. The study also demonstrates a simple text-driven image editing application, suggesting that edits are most effective when CIS is between 0.5 and 0.7.

**Strengths:**

1. Exploring when concepts emerge along the diffusion timeline offers an interesting and novel temporal perspective on concept formation in generative models.

2. The paper is clearly written and easy to follow, with well-structured figures and explanations that make the methodology and findings accessible even to non-experts.

**Weaknesses:**

1. Narrow task scope and unclear practical significance (main concern).
The paper leverages PCI and CIS to reveal that timing, model choice, and context influence concept insertion, but the exploration remains confined to a single downstream application—text-driven image editing—and primarily relies on qualitative demonstrations. Section 5 and Figure 5 (as well as Appendix Figure 11) merely show edits guided by CIS thresholds and report the empirical finding that the 0.5–0.7 probability range offers a “balanced trade-off.” However, the study does not substantiate why knowing that certain concepts lock in earlier or later yields quantifiable benefits in practice or theory—such as reducing trial-and-error, improving editing stability, strengthening concept binding, or informing training/scheduler design. Moreover, focusing exclusively on the [0.5, 0.7] CIS window without connecting it to broader temporal or theoretical frameworks limits the work’s contribution to understanding “when timing matters” beyond this heuristic range.

2. Strong reliance on a single LVLM-based VQA judge with potential coupling bias.
The CIS measure depends entirely on a single VQA-style Large Vision–Language Model (Qwen-VL-3B) for concept detection. Although Appendix A.4.1 includes a brief comparison with a 7B variant, there is no human annotation calibration or multi-judge agreement analysis. The evaluation also remains binary (concept present/absent), which cannot capture concept strength, spatial accuracy, or attribute binding, potentially conflating weak or erroneous detections with genuine insertions. These factors introduce uncertainty into the CIS curve and may distort the estimated locking points.

3. Editing experiments lack strong baselines and quantitative comparison.

**Questions:**

See weakness1.

---

> ### Author Response · Authors · 2025-11-21
> **Author response to Reviewer uBmP (Part 1/3)**
>
> We thank reviewer uBmP for their constructive feedback on our work. Below, we address each of the concerns raised.
>
> **(W1. narrow task scope and unclear practical significance...)**
>
> We thank the reviewer for raising this concern. We have addressed this in four complementary ways:
>
> **1.1**. Following the reviewer’s feedback, we would first like to clarify that our central contribution is the **temporal analysis** framework and the **novel empirical findings on concept emergence** and stabilization rather than proposing a new editing method. The editing task is intended as a *proof-of-concept downstream use case*, illustrating how these insights can be operationalized beyond discovering, for example, model-inherent biases.
>
> **1.2**. **Demonstrated Practical Benefit via New Quantitative Evaluation.**
> To address the request for concrete evidence of usefulness, we have added a new quantitative evaluation (Section 5, Table 1; see response to W3). We evaluate edited images using three complementary CLIP-based metrics that are widely used in the editing literature [1]:
> - **$\mathbf{CLIP}_{img}$** globally measures the perceptual similarity between the base image and the edited image in the CLIP image-embedding space. This captures overall content preservation, ensuring that changes do not unintentionally alter unrelated aspects of the scene.
>
> - **$\mathbf{CLIP}_{txt}$** globally measures the alignment between the edited image and the concept prompt in the CLIP text–image embedding space. This captures the global semantic strength of the inserted concept.
>
> - **$\mathbf{CLIP}_{dir}$** compares the image-edit direction (base → edited image) with the textual direction (base prompt → concept prompt). Rather than isolating the concept, this metric evaluates whether the semantic shift produced by the edit is consistent with the semantic shift implied by the prompts. High $\mathbf{CLIP}_{dir}$ indicates that the deviation introduced by editing is aligned with the intended conceptual change rather than being an arbitrary drift.
>
> Together, these three metrics provide a comprehensive quantification of the insertion–preservation trade-off. The obtained results are coherent with the temporal findings reported in the main analysis. Specifically, when we perform the editing experiment across increasing intervention CIS bands, we observe a systematic decrease in **$\mathbf{CLIP}_{img}$**, indicating reduced content preservation as the edit is applied earlier in the denoising process. At the same time, both **$\mathbf{CLIP}_{txt}$** and **$\mathbf{CLIP}_{dir}$** increase with increasing CIS band, reflecting stronger alignment with the target concept and a more consistent semantic shift relative to the intended prompt change. This temporal trend mirrors the insertion dynamics observed in our CIS curves: early interventions produce stronger concept insertion but at the cost of larger deviation from the base image.
>
> The quantitative results (Section 5, Table 1, and provided below) further highlight the advantages of our CIS-guided strategy over strong editing baselines NTI+P2P [2] and Stable Flow [1]. Our editing window of **[0.5, 0.7]**, which we identified through our CIS-based analysis, consistently provides the best balance compared to NTI+P2P and Stable Flow, achieving meaningful concept edits while maintaining high fidelity to the original image.
>
> | **Method**          | $\mathbf{CLIP}_{img}$ | $\mathbf{CLIP}_{txt}$ | $\mathbf{CLIP}_{dir}$ |
> |---------------------|--------------|--------------|--------------|
> | NTI+P2P             | 0.8666       | 0.2215       | 0.0979       |
> | Stable Flow         | 0.8324   | 0.2152       | 0.0631       |
> | **PCI-τ$_{30}$**         | 0.9343       | 0.2125       | 0.1014       |
> | **PCI-τ$_{50}$**         | **0.8885**   | 0.2236       | 0.1387       |
> | **PCI-τ$_{60}$**         | 0.8625       | 0.2289       | 0.1531       |
> | **PCI-τ$_{70}$**         | 0.8353       | **0.2341**   | **0.1678**   |
> | **PCI-τ$_{90}$**         | 0.7679       | 0.2449       | 0.1963       |

---

> ### Author Response · Authors · 2025-11-21
> **Author response to Reviewer uBmP (Part 2/3)**
>
> **(W1. narrow task scope and unclear practical significance...)**
>
> **1.3.** **Broadened and Strengthened Experimental Scope.**
> We have expanded our study beyond a single setting by including:
> - **additional generative models** (PixArt-α [3], FLUX [4]) ,
>
>  - **reverse experiments** on concept deletion (Appendix A.2),
>
> - **compositional prompts** with multiple interacting concepts (Appendix A.7.3),
>
> showing that **temporal effects generalize across architectures, tasks, and semantic structures**.
>
> **1.4.** Importantly, we now also clarify why our temporal findings have practical value for editing pipelines. Today’s diffusion editing workflows often rely on heuristic trial-and-error (e.g., “adjust the strength,” “inject earlier,” “inject later”), which is both slow and unpredictable. Our analysis explicitly identifies *when* semantic updates are maximally impactful yet minimally disruptive along the denoising trajectory. This provides a principled alternative to manual tuning.
>
> Our new quantitative results show that choosing the timestep according to CIS leads to fewer failed edits, more stable concept binding, and significantly reduced unintended drift. Finally, from a theoretical perspective, our work offers the first systematic quantification of concept fixation timing across diverse models. This goes beyond the standard “early structure / late detail” intuition and provides a more precise, concept-level characterization of diffusion dynamics, contributing to the broader understanding of when timing matters in generative models.
>
> ---
>
> **(W2. multi-judge agreemenet and binary evaluation...)**
>
> We have addressed these concerns in two ways:
>
> 1. **Multi-Judge Agreement.**
> To ensure our results are not an artifact of a single LVLM, we have **expanded our VQA evaluation to include two additional models: LLaVA-OneVision-7B [5] and SmolVLM-2B [6]**. We show CIS curves using all four models (QwenVL-3B, QwenVL-7B, LLaVA-OneVision-7B, SmolVLM-2B) in Appendix A.5.1 and find that the observed trends and relative concept locking points are highly similar. This new analysis demonstrates that our findings are stable across different VQA judges.
>
> 2. **Binary Evaluation.**
> Thank you for raising this interesting point. While our CIS metric is indeed binary, your comment motivated us to conduct an additional, qualitative cross-attention–based analysis (Appendix A.7.2). This analysis examines token-level cross-attention maps for the concept words across different CIS probabilities, allowing us to explore how concepts tend to manifest spatially in the generated images. Moreover, these cross-attention maps provide a continuous measure of concept appearance by examining the attention strength across time.
>
>    The visual patterns suggest that higher-CIS interventions often correspond to clearer and more localized cross-attention activations on concept-relevant regions. This provides an intuitive, spatially grounded view of concept insertion that complements our binary score and offers additional insight into how concept insertability evolves over the denoising trajectory. Importantly, we emphasize that this analysis is qualitative and meant to contextualize rather than replace the main binary evaluation. Attention-based methods work well for simple prompts but require careful adjustment and normalization [7,8]; for semantically richer concepts, a strong reasoning model is needed, which is why we opted for the VLM with binary answers.
>
> ---
>
> **(W3. editing experiments lack strong baselines...)**
>
> To address this, we have added a **new quantitative evaluation** (Section 5, Table Y) for our editing application. For this experiment, we generated a **dedicated evaluation dataset** consisting of **88 concepts**, each associated with a base prompt and a corresponding concept prompt. For every concept, we sampled **20 random seeds** for generating the images through SDXL, yielding a diverse and balanced dataset for scoring. We evaluate our method alongside SOTA editing approaches, NTI+P2P [2] and Stable-Flow [1]. All methods, our CIS-guided approach (with SDXL), NTI+P2P (with SD-1.5), and Stable-Flow (with FLUX), were evaluated on this exact dataset to ensure a fair and controlled comparison.
> We discussed the results above in our answer to your raised Weakness W1 as we believe it further emphasizes the practical significance of our work. In brief, our CIS-guided editing window of **[0.5, 0.7]** consistently provides the best balance compared to NTI+P2P and Stable Flow, achieving meaningful concept edits while maintaining high fidelity to the original image.

---

> ### Author Response · Authors · 2025-11-21
> **Author response to Reviewer uBmP (Part 3/3)**
>
> **References**
>
> [1] O. Avrahami, O. Patashnik, O. Fried, E. Nemchinov, K. Aberman, D. Lischinski, and D. Cohen-Or, “Stable Flow: Vital Layers for Training-Free Image Editing,” in *Proc. IEEE/CVF Conf. on Computer Vision and Pattern Recognition (CVPR)*, 2025.
>
> [2] R. Mokady, A. Hertz, K. Aberman, Y. Pritch, and D. Cohen-Or, “Null-Text Inversion for Editing Real Images using Guided Diffusion Models,” in *Proc. IEEE/CVF Conf. on Computer Vision and Pattern Recognition (CVPR)*, 2023.
>
> [3] J. Chen et al., “PixArt-α: Fast Training of Diffusion Transformer for Photorealistic Text-to-Image Synthesis,” *arXiv preprint arXiv:2310.00426*, 2023.
>
> [4] Black Forest Labs, “FLUX,” 2024. Online: https://github.com/black-forest-labs/flux
>
> [5] Y. Chen, Y.-X. Wang, D.-A. Huang, L.-J. Li, R. Krishna, and Y. Xu, “LLaVA-OneVision: Easy Visual Task Transfer,” *arXiv preprint arXiv:2407.13531*, 2024.
>
> [6] T. A. Adediran and S. Al-Dabagh, “SmolVLM: Redefining small and efficient multimodal models,” *arXiv preprint arXiv:2405.09334*, 2024.
>
> [7] A. Helbling, T. H. S. Meral, B. Hoover, P. Yanardag and D. H. Chau, “ConceptAttention: Diffusion Transformers Learn Highly Interpretable Features,” in Proc. 42nd International Conference on Machine Learning (ICML), 2025.
>
> [8] A. Kukleva, E. Simsar, A. Tonioni, M. F. Naeem, F. Tombari, J. E. Lenssen and B. Schiele, “RefAM: Attention Magnets for Zero-Shot Referral Segmentation,” *arXiv preprint arXiv:2509.22650*, 2025.

---

> > ### Comment · Reviewer_uBmP · 2025-11-22
> > **Resbonse to authors**
> >
> > Thank you for the authors’ additional experiments and analyses. However, the appendix now contains too many experimental settings and results, and the organization of some sections remains confusing (for example, CDS is introduced within the CIS section). This makes the paper difficult to follow. It would be better to reorganize the manuscript as a whole, rather than simply inserting extra information into the current version.
> >
> > In addition, is it still appropriate to keep “CIS” in the title? A more general title and a broader writing perspective might further strengthen the work.

---

> > > ### Author Response · Authors · 2025-11-26
> > > **Author response to Reviewer uBmP**
> > >
> > > We sincerely thank Reviewer uBmP for this helpful observation. We agree that the previous Appendix structure mixed experimental settings, ablations, and supplementary analyses in a way that made navigation difficult. In the revised version, we have **fully reorganized** the Appendix into clearly separated sections, and **ensured** that CDS is presented only in its own dedicated subsection, avoiding any confusion with CIS. We believe this substantially improves clarity.
> > >
> > > We also appreciate the reviewer’s comment regarding the title. The earlier framing placed too much emphasis on CIS, whereas the paper now presents a broader investigation of temporal concept behaviour through PCI. Following this suggestion, we have updated the title to the more general and representative:
> > >
> > > “Temporal Concept Dynamics in Diffusion Models via Prompt-Conditioned Interventions”
> > >
> > > In addition, we have revised the introduction and method to better reflect this broader perspective, clarify the positioning of CIS and CDS, and improve the overall narrative.
> > > We thank the reviewer for these valuable recommendations, which have strengthened the clarity, organization, and presentation of the paper.

---

### Official Review · Reviewer_Qc2X · 2025-10-30

**Soundness:** 3
**Presentation:** 3
**Contribution:** 3
**Rating:** 8
**Confidence:** 4

**Summary:**

This paper introduces PCI, a training-free framework for analyzing when concepts become locked into the generation trajectory of text-to-image diffusion models. The core method involves switching prompts at different timesteps during denoising: starting with a neutral base prompt and then introducing a concept-augmented prompt at various intervention points. The authors define CIS as the probability that a concept inserted at a given timestep appears in the final image, evaluated using a VLM.

**Strengths:**

- PCI is training-free, model-agnostic, and requires no access to model internals, making it broadly applicable and easy to implement across different diffusion architectures.
- The seed resampling strategy with optional negative guidance ensures that base prompts remain neutral with respect to target concepts, addressing a potential confound that could undermine the analysis.
- While the core idea of prompt switching is conceptually straightforward, the paper addresses a dimension that prior interpretability work has largely overlooked: the temporal evolution of concepts during generation. Through thorough experimentation across diverse concept taxonomies, the study reveals actionable patterns (e.g., global factors lock early, human properties mid-trajectory, accessories late) that provide valuable insights, filling an important gap in understanding diffusion model dynamics.

**Weaknesses:**

- The paper primarily focuses on CIS as the main metric for evaluating when concepts can be inserted. However, successful concept insertion does not necessarily mean the generated image maintains fidelity to the original intent or preserves other important content from the base prompt. The trade-off between concept insertion success and overall content preservation is not systematically quantified beyond qualitative observations in the editing examples. A more comprehensive analysis could include metrics that measure how much the image deviates from the base generation when concepts are inserted at different timesteps, helping users better understand the full cost-benefit landscape of intervention timing.
- The prompts studied in this work are relatively simple and focused (e.g., "a realistic photo of a person," "a landscape"). How well do the insights and CIS patterns generalize to more complex, compositional prompts with multiple objects, relationships, and attributes? For instance, if editing a complex scene like "a young woman wearing sunglasses sitting on a red chair in a sunlit café," would the optimal insertion times for individual concepts (age, accessories, color, setting) still follow the patterns observed with simpler prompts, or would the interactions between multiple concepts shift the temporal dynamics?

**Questions:**

- if the inserted concept is very abstract and cannot be easily fit in the categories studied in this work, how can a user find the best inserting point without trying every possible timestep?
- what if the VLM cannot evaluate the abstract concept, how do you solve this? Could you discuss failure modes where VQA might struggle (e.g., highly subjective concepts like "elegance" or "tranquility") and potential mitigation strategies?
- The current analysis focuses on inserting one concept at a time. How would CIS behave when multiple concepts need to be inserted? Would their optimal insertion times interfere with each other?

---

> ### Author Response · Authors · 2025-11-21
> **Author response to Reviewer Qc2X (Part 1/2)**
>
> We thank reviewer Qc2X for their constructive feedback on our work. Below, we address each of the concerns raised.
>
> **(W1. trade-off between concept insertion success and overall content preservation...)**
>
> Thank you for this constructive and helpful feedback. To address this limitation, we have added a new quantitative evaluation of our editing application (Section 5, Table 1), explicitly measuring the trade-off between concept insertion success and overall content preservation. Specifically, we evaluate each edited image using three complementary CLIP-based metrics suggested in the editing literature [1]:
>
> - **$\mathbf{CLIP}_{img}$** globally measures the perceptual similarity between the base image and the edited image in the CLIP image-embedding space. This captures overall content preservation, ensuring that changes do not unintentionally alter unrelated aspects of the scene.
>
> - **$\mathbf{CLIP}_{txt}$** globally measures the alignment between the edited image and the concept prompt in the CLIP text–image embedding space. This captures the global semantic strength of the inserted concept.
>
> - **$\mathbf{CLIP}_{dir}$** compares the image-edit direction (base → edited image) with the textual direction (base prompt → concept prompt). Rather than isolating the concept, this metric evaluates whether the semantic shift produced by the edit is consistent with the semantic shift implied by the prompts. High **$\mathbf{CLIP}_{dir}$** indicates that the deviation introduced by editing is aligned with the intended conceptual change rather than being an arbitrary drift.
>
> Together, these three metrics provide a more complete quantification of the insertion–preservation trade-off. Our results are also coherent with the temporal findings reported in the main analysis.
> Specifically, when we perform the editing experiment across increasing intervention CIS bands, we observe a systematic decrease in **$\mathbf{CLIP}_{img}$**, indicating reduced content preservation as the edit is applied earlier in the denoising process. At the same time, both **$\mathbf{CLIP}_{txt}$** and **$\mathbf{CLIP}_{dir}$** increase with increasing CIS band, reflecting stronger alignment with the target concept and a more consistent semantic shift relative to the intended prompt change. This temporal trend mirrors the insertion dynamics observed in our CIS curves: early interventions produce stronger concept insertion but at the cost of larger deviation from the base image.
>
> The quantitative results (Section 5, Table 1, and also provided below) further highlight the advantages of our CIS-guided strategy over strong editing baselines NTI+P2P [2] and Stable Flow [1]. Our CIS-guided editing window of **[0.5, 0.7]** consistently provides the best balance compared to NTI+P2P and Stable Flow, achieving meaningful concept edits while maintaining high fidelity to the original image.
>
> | **Method**          | $\mathbf{CLIP}_{img}$ | $\mathbf{CLIP}_{txt}$ | $\mathbf{CLIP}_{dir}$ |
> |---------------------|--------------|--------------|--------------|
> | NTI+P2P             | 0.8666       | 0.2215       | 0.0979       |
> | Stable Flow         | 0.8324   | 0.2152       | 0.0631       |
> | **PCI-τ$_{30}$**         | 0.9343       | 0.2125       | 0.1014       |
> | **PCI-τ$_{50}$**         | **0.8885**   | 0.2236       | 0.1387       |
> | **PCI-τ$_{60}$**         | 0.8625       | 0.2289       | 0.1531       |
> | **PCI-τ$_{70}$**         | 0.8353       | **0.2341**   | **0.1678**   |
> | **PCI-τ$_{90}$**         | 0.7679       | 0.2449       | 0.1963       |
>
> ---

---

> ### Author Response · Authors · 2025-11-21
> **Author response to Reviewer Qc2X (Part 2/2)**
>
> **(W2. generalization to multiple concepts and complex prompts...)**
>
> Thank you for this constructive question. Our method is explicitly designed to analyze the temporal insertability of concepts regardless of prompt length or structure, since PCI intervenes by switching only the specific concept-bearing tokens while keeping the remaining context unchanged. To evaluate whether our insights extend beyond simple prompts, we conducted additional multi-concept experiments (Appendix A.7.3), where compositional concept prompts (e.g., female + sketch, old + Asian) are inserted jointly.
>
> These new results show that for many concept pairs, we observe that the resulting CIS curves qualitatively match the individual single-concept insertion trends. In these cases, the multi-concept CIS behaves as a near-compositional combination of the two single-concept dynamics, indicating weak or neutral interaction between the concepts. Representative examples include pairs such as *coat + snowy* and *female + Asian*, where the temporal insertion windows remain similar to the single-concept baselines.
>
> However, some concept combinations exhibit markedly different temporal behavior compared to their single-concept counterparts. Notable examples include *female + sketch*. For this pair, we observe that the effective insertion window shifts significantly further into later timesteps, with high insertion success maintained much deeper in the denoising trajectory than for either concept alone. This is particularly interesting because certain concept types, especially styles such as *sketch*, are typically inserted very early in the trajectory. The fact that joint insertion remains possible at later timesteps suggests that specific combinations may influence the model’s temporal allocation of representational capacity.
>
> These findings reveal that concept interactions are not uniformly additive. Instead, diffusion models exhibit non-linear compositional behavior: some concept pairs behave independently, while others interact in ways that reshape their temporal insertion windows. Understanding these interactions is essential for predicting the feasibility and stability of multi-attribute text-driven image editing and highlights the importance of PCI as a diagnostic tool beyond the single-concept setting. All of the corresponding discussion is now included in Appendix A.7.3.
>
> ---
>
> **(Q1 and Q2. abstract concepts...)**
>
> We thank the reviewer for sparking this discussion. We agree that abstract concepts can pose a challenge for VLMs and that such specific settings might influence our results. We have conducted an additional analysis on three abstract concepts: *elegant*, *art*, and *creative*. These concepts are inherently subjective and their interpretation may vary across cultures and viewpoints. To better understand how the VQA model behaves in this setting, we computed CIS curves for these three concepts (see Appendix A.7.4). The raw results show that the VLM still produces smooth CIS curves, but with noticeably different shapes: for instance, the CIS curve for *creative* and *art* starts with a probability of around 0.8 and drops below 0.5 much earlier in the timestep sequence than for *elegant*. This suggests that the VLM has its own idea of what “elegant,” “art,” or “creative” mean, likely stemming from its training data.
>
> To account for subjectivity of individual models, we could perform multi-model agreement. Specifically, we can aggregate CIS curves from multiple VLMs (trained on different data) to calibrate the CIS curves. We have added a section on multi-model agreement of four VLMs in Appendix A.5.1. If subjective ratings of an abstract concept are desired, we could instead extend the prompt to reflect the subjectiveness. For example, in the prompt “In Eastern Asia, would this image be considered elegant?”, *Eastern Asia* provides the desired subjective context in which a user might want to evaluate CIS.
>
> ---
>
> **(Q3. multiple concept interaction...)**
>
> We appreciate the reviewer’s question also raised in their Weakness W2, where we adressed this concern.
>
> ---
> **References**
>
> [1] O. Avrahami, O. Patashnik, O. Fried, E. Nemchinov, K. Aberman, D. Lischinski, and D. Cohen-Or, “Stable Flow: Vital Layers for Training-Free Image Editing,” in *Proc. IEEE/CVF Conf. on Computer Vision and Pattern Recognition (CVPR)*, 2025.
>
> [2] R. Mokady, A. Hertz, K. Aberman, Y. Pritch, and D. Cohen-Or, “Null-Text Inversion for Editing Real Images using Guided Diffusion Models,” in *Proc. IEEE/CVF Conf. on Computer Vision and Pattern Recognition (CVPR)*, 2023.

---

> > ### Comment · Reviewer_Qc2X · 2025-11-22
> >
> > I thank the authors for the detailed response. I have two follow-up questions:
> > 1. I'm not sure if I've interpreted Figure 15 correctly. For each concept composition, you compare the CIS curve of inserting a single concept versus jointly inserting two concepts. However, I notice that the "multi" curves in the left and right plots appear different. Shouldn't the jointly inserted curves in both subplots be identical, since they represent the same multi-concept insertion?
> > 2. Out of curiosity regarding abstract concept insertion, based on the experimental results, the authors suggest in lines 1245-1246 that "This suggests that the VLM has its own idea of what “elegant”, “art”, or “creative” means, which likely stems from its training data." In Figure 10, you performed an ablation study across different LVLMs and concluded that they produce closely matching CIS trajectories. Does this finding also hold for these three abstract concepts? It's possible that each model has its own interpretation of these concepts due to its training data, but this might not significantly affect the CIS curves themselves. If so, we could still interpret the results globally even for abstract concepts, which I think would be a really interesting finding.

---

> > > ### Author Response · Authors · 2025-11-26
> > > **Author response to Reviewer Qc2X**
> > >
> > > We sincerely thank Reviewer Qc2X for staying engaged in the discussion. Below, we address each of the questions raised.
> > >
> > > **Q1.**
> > >
> > > Thank you for this question. We realize that the original description of the multi-concept plots may not have made the evaluation procedure fully clear, and we have now clarified this more explicitly in the revised Appendix.
> > >
> > > In particular, CIS is a concept-specific rather metric than prompt-specific. Even when the prompt contains multiple concepts (e.g., old + Asian), the CIS computation evaluates the insertion success of **one target concept at a time**. Therefore, each subfigure in Figure 15 (now updated as Figure D4) contains two different CIS evaluations. For example, in Figure D4-a:
> > >
> > > - **Left plot:** CIS(concept *Asian*) is being measured through the VQA model for:
> > >   - single-concept prompt: “an image of an *Asian* person”
> > >   - multi-concept prompt: “an image of an *old Asian* person”
> > >
> > > - **Right plot:** CIS(concept *old*) is being measured through the VQA model for:
> > >   - single-concept prompt: “an image of an *old* person”
> > >   - multi-concept prompt: “an image of an *old Asian* person”
> > >
> > > Although both subplots use the same multi-concept prompt, the target concept being measured differs, so the resulting CIS curves also differ. Concretely, given concept A and concept B:
> > >
> > > - the left subplot shows CIS(A | A+B prompt) vs CIS(A | A prompt)
> > > - the right subplot shows CIS(B | A+B prompt) vs CIS(B | B prompt)
> > >
> > > We specifically kept the CIS curves single-concept to still be able to reflect when a model starts introducing or locking individual features. Despite having a multi-concept prompt, still it could be that the first concept (e.g. “old”) is locked in while the second concept is still changeable (e.g. “Asian”). We have updated the text in the revision to make this evaluation protocol clearer. In case there are remaining issues, please let us know.
> > >
> > > **Q2.**
> > >
> > > We thank the reviewer for this insightful question. Following your suggestion, we examined the CIS trajectories for the three abstract concepts {*art, creative, elegant*} **across all four LVLMs** used in our study. The corresponding results are shown in Figure D5, and we have added the following discussion to Appendix Sec. D.4.
> > >
> > > Across all three concepts, the three larger models (Qwen-3B, Qwen-7B, and LLaVA-OneVision-7B) display similar CIS dynamics: their trajectories are closely aligned throughout the entire denoising process, and their characteristic decay profiles nearly coincide. In contrast, **SmolVLM-2B consistently produces lower CIS values**. Its curves start from a much lower initial success rate and drop more sharply, indicating a significantly weaker ability to detect or validate these abstract attributes when inserted late in the trajectory.
> > >
> > > Despite this degradation in absolute performance, SmolVLM-2B still follows the *same qualitative trend*, a smooth, monotonic decrease from early to late timesteps, suggesting that its notion of concept insertion is directionally aligned with the larger models but less reliable in magnitude.
> > >
> > > Taken together, these observations indicate that abstract concepts such as “art,” “creative,” and “elegant” are interpreted in a consistent way across strong LVLMs, with only the smallest model deviating in strength but not in pattern. This supports our claim that CIS-based temporal conclusions generalize robustly across models and hold not only for concrete attributes but equally for high-level, abstract ones.

---

### Official Review · Reviewer_V3XN · 2025-10-31

**Soundness:** 3
**Presentation:** 4
**Contribution:** 3
**Rating:** 6
**Confidence:** 4

**Summary:**

This paper studies how individual concepts in a text prompt influence the diffusion generation process over time. The proposed method, Prompt-Conditioned Intervention (PCI), provides an intuitive mechanism to probe when a specific concept meaningfully affects generation. In addition, the paper introduces the Concept Insertion Success (CIS) metric to quantitatively capture the temporal dynamics of concept influence during diffusion. Together, PCI and CIS offer a useful framework for analyzing how textual concepts guide the image generation trajectory.

**Strengths:**

- The idea behind PCI and CIS are intuitive and well-motivated, offering a practical way to examine concept influence in diffusion models.
- The manuscript is clearly written and the main claims are clearly communicated.

**Weaknesses:**

- The analysis is primarily focuses on isolated concept influence (while the interactions between concepts and contexts are mentioned). Further exploration of interactions between concepts and broader prompt context would enrich the contributions.
- While PCI and CIS measure the latest timestep at which a concept can be successfully inserted, this does not directly indicate when the model begins to encode the concept. Studying the earliest insertion or concept disappearance behavior would strengthen the temporal interpretation of the findings.
- Sensitivity analysis to variations in prompt phrasing and initial noise seeds is limited (as it is averaged over subcategory of concepts). Investigating robustness across seeds and prompt variations would help clarify the generality of PCI/CIS.

**Questions:**

In addition to insertion, could PCI and CIS be extended to analyze the concept removal or replacement dynamics? For example, identifying the latest timestep at which removing a concept no longer changes the generated output.

---

> ### Author Response · Authors · 2025-11-21
> **Author response to Reviewer V3XN**
>
> We thank reviewer V3XN for their constructive feedback on our work. Below, we address each of the concerns raised.
>
> **(W1. multiple concept interaction...)**
>
> We thank the reviewer for this interesting suggestion. We have expanded our analysis to include multi-concept PCI experiments, where two concepts are inserted jointly. These new results, now included in Appendix A.7.3, show that for many concept pairs, we observe that the resulting CIS curves qualitatively match the individual single-concept insertion trends. In these cases, the multi-concept CIS behaves as a near-compositional combination of the two single-concept dynamics, indicating weak or neutral interaction between the concepts. Representative examples include pairs such as *coat + snowy* and *female + Asian*, where the temporal insertion windows remain similar to the single-concept baselines.
>
> However, some concept combinations exhibit markedly different temporal behavior compared to their single-concept counterparts. Notable examples include *female + sketch*. For this pair, we observe that the effective insertion success maintained much deeper in the denoising trajectory than for either concept alone. This is particularly interesting because certain concept types, especially styles such as *sketch*, are typically inserted very early in the trajectory. The fact that joint insertion remains possible at later timesteps suggests that specific combinations may influence the model’s temporal allocation of representational capacity.
>
> These findings reveal that concept interactions are not uniformly additive. Instead, diffusion models exhibit non-linear compositional behavior: some concept pairs behave independently, while others interact in ways that reshape their temporal insertion windows. Understanding these interactions is essential for predicting the feasibility and stability of multi-attribute text-driven image editing and highlights the importance of PCI as a diagnostic tool beyond the single-concept setting. All of the corresponding discussion is now included in Appendix A.7.3.
>
> ---
>
> **(W2. concept disappearance behavior...)**
>
> Thank you for this constructive suggestion. To provide a complementary temporal view, we have **conducted a new set of experiments on concept deletion (or "disappearance")**. In this new analysis (added to Appendix A.2), we start with a prompt containing the concept (e.g., "a photo of a **red** car") and *remove* the concept (here, "red") at different timesteps. We then measure the probability that the concept "red" *still appears* in the final image. This "reverse-CIS" analysis helps identify when a concept becomes "un-removable" and its encoding is finalized, strengthening our temporal interpretation as suggested.
>
> Based on our results in Appendix A.2, we observe that deletion dynamics behave *differently* from insertion dynamics, but not in a universally uniform manner. While CIS curves exhibit substantial variability in when different concepts become insertable, concept deletion success (CDS) curves reveal a distinct trend: concept deletion tends to fail considerably earlier in the denoising trajectory. However, there is still noticeable variation across concepts, categories, and prompt configurations, indicating that deletion is not governed by a single universal threshold.
>
> Two factors help explain this behaviour. First, when generation begins under the concept prompt, the model is effectively primed toward that concept from the earliest steps, making subsequent removal more difficult and often shifting the effective deletion window to earlier timesteps. Second, certain concept pairs (e.g., binary attribute switches such as *night* vs. *morning*) behave differently from cases where a concept is simply present or absent, leading to sharper or more gradual transition behaviours depending on semantic structure.
>
> ---
>
> **(W3. sensitivity analysis...)**
>
> We apologize for the lack of clarity. Our original experiments **were already averaged over 100 seeds** for each concept, and we included an ablation on this in Appendix A.5.2. We have now emphasized this more clearly in the main text (Section 4, lines 254–255). Furthermore, to explicitly address **prompt sensitivity**, we have added a new analysis (Appendix A.5.3). We test five different semantic paraphrasings for three different subcategories. The results show that while the exact CIS curve can shift slightly, the relative temporal ordering of concept types (e.g., "actions" are inserted earlier than "objects") remains highly consistent, demonstrating the robustness of our findings to prompt phrasing.
>
> ---
>
> **(Q1. concept removal...)**
>
> We appreciate the reviewer’s question also raised in their Weakness W2, where we adressed this concern.

---

### Official Review · Reviewer_RMvL · 2025-10-31

**Soundness:** 3
**Presentation:** 3
**Contribution:** 2
**Rating:** 4
**Confidence:** 4

**Summary:**

This paper investigates the denosing dynamics of concept formation in diffusion models, focusing on when specific concepts (e.g., age) emerge and stabilize during the denoising trajectory. It proposes PCI (Prompt-Conditioned Intervention), a training-free and model-agnostic framework that analyzes Concept Insertion Success (CIS) to study how concepts evolve over diffusion time.  Experiments on state-of-the-art text-to-image diffusion models reveal some insights about diverse temporal behaviors for concept formation.

**Strengths:**

- The presentation of figures is great and easy to understand.
- The math notations in this paper are self-contained and well-defined.
- The paper writing is easy to follow.
- The idea of Prompt-Conditioned Intervention is cool.
- The proposed method is straightforward.

**Weaknesses:**

- I am quite disappointed that, although this paper uncovers some interesting phenomena regarding denoising trajectories, the final method does not stand out significantly. This is my main concern.
- PCI is also heavily based on the performance of the adopted MLLM (like Qwen-3B).
- The evaluation is only based on SD-series models. How about FLUX and other SoTA models?
- Some wrong citation formats are used in the paper.
- I have to say that the performance of image editing is not so well in Fig. 5. It seems that the resultant methods didn't work well.

**Questions:**

Please see the section of weaknesses.

---

> ### Author Response · Authors · 2025-11-21
> **Author response to Reviewer RMvL (Part 1/2)**
>
> We thank reviewer RMvL for their constructive feedback on our work. Below, we address each of the concerns raised.
>
> **(W1. limited significance or contribution of the final method...)**
>
> We thank the reviewer for sharing their concern. We would like to clarify that the primary goal of our work is the *temporal analysis of concept emergence* in diffusion models, which to the best of our knowledge has not been achieved in this scope. We agree that our initial experiments regarding editing as a direct down-stream application, where our insights fell a bit short in terms of usefullnes, which is why we now extended its scope significantly.
>
> First, we  provide a new large-scale **quantitative evaluation** of our editing application (Section 5, Table 1) using 1,760 edited images across 88 concepts. We rely on established CLIP-based metrics commonly used for evaluation in the editing literature [1] (**$\mathbf{CLIP}_{img}$**, **$\mathbf{CLIP}_{txt}$**, **$\mathbf{CLIP}_{dir}$**). we compare our CIS-guided strategy to two strong editing baselines, NTI+P2P [2] and Stable Flow [1]. Across these metrics, our method achieves the most favorable balance between concept insertion and preservation of the original content, demonstrating that **PCI provides a principled mechanism for choosing effective and minimally disruptive intervention windows**. We additionally include qualitative comparisons in Appendix A.8, showing that CIS-guided edits are often more targeted and introduce less unintended drift than both baselines.
>
> Motivated by this comment, we also expanded the breadth and depth of the core PCI analysis. We now include new experiments on (i) **VLM diversity**, adding LLaVA-OneVision-7B [3] and SmolVLM-2B [4] to confirm that CIS curves are consistent across four independent judges (Appendix A.5.1); (ii) **model diversity**, adding PixArt-α [5] and FLUX.1-dev [6] to show that temporal insertability patterns persist across DiT-, UNet-, and rectified-flow architectures; (iii) **concept deletion**, where we remove a concept at timestep *t* to measure when it becomes impossible to erase, a complementary “reverse” perspective that strengthens the temporal interpretation (Appendix A.2); (iv) **multi-concept interactions**, showing that certain concept pairs display non-linear temporal behaviors, not predictable from single-concept curves (Appendix A.7.3); and (v) **prompt robustness**, where we evaluate five paraphrased prompts per concept and find that the temporal ordering of concept types remains stable (Appendix A.5.3). We further provide a qualitative cross-attention analysis (Appendix A.7.2) to complement the binary CIS score with a spatial and continuous perspective, revealing how the attention patterns of concept tokens evolve across different CIS bands.
>
> Taken together, these additions strengthen both the scientific insights and practical significance for down-stream use cases of our analysis.Thanks to the constructive feedback that we received during the rebuttal, our updated manuscript more clearly demonstrates both the novelty and the practical value of our contributions.
>
> ---
>
> **(W2. limited LVLM diversity..).**
>
> To ensure our results are not an artifact of a single LVLM, we now have **expanded our VQA evaluation to include two additional models: LLaVA-OneVision-7B [3] and SmolVLM-2B [4]**. We show CIS curves of all four models (Qwen-3B, Qwen-7B, LLaVA, SmolVLM) in Appendix A.5.1 and find that the observed trends and relative concept locking points are highly similar. This new analysis suggests that our findings are stable and robust across different VQA models.
>
> ---
>
> **(W3. evaluation scope limited to SD-series models...)**
>
> Thank you for this suggestion. We agree that expanding the model variety strengthens our claims. We now **added new experiments on two additional SOTA models: PixArt-α XL [5]** (a diffusion-based transformer (DiT) model) and **FLUX.1-dev [6]**. We have updated Figs. 3 and 12 to include results for FLUX and PixArt-α, and we have revised the corresponding discussions in Section 4.2 (e.g., line 352) to incorporate these models.
>
> The additional results confirm that the overall CIS behavior is consistent across architectures: **PixArt-α (DiT-based diffusion model) shows the same smooth, monotonic CIS curves and similar concept-locking intervals as UNet-based models**. Separately, we also observe **strong coherence between the two rectified-flow models, SD 3.5 and FLUX**, which produce similar CIS curves with earlier insertion points across concepts.

---

> ### Author Response · Authors · 2025-11-21
> **Author response to Reviewer RMvL (Part 2/2)**
>
> **(W4. citation formats...)**
>
> We sincerely apologize for these formatting errors. We have carefully reviewed the entire manuscript and corrected citations. In case there are remaining issues, please let us know.
>
> ---
>
> **(W5. image editing quality...)**
>
> We are sorry for the confusion regarding Figure 5. To clarify, the *central two columns* are the range where insertion is just still possible *suggested by our analysis*, i.e., where we would expect good editing quality. And this is indeed the case, the results show that for these time points we receive good generation where the target concept is visible while maintaining the structure of the original image.
> To better guide the reader, we adapted the presentation in Figure 5. We explicitly showed other values of insertion times (different τ) to make clear that knowing the time when insertion is still possible --- exactly what our analysis gives us --- is crucial.
>
> To subtantiate our claims further, we present an additional set of qualitative examples (Appendix A.8) and a new *quantitative* analysis of editing quality (Sec. 5, Table 1, and provided below), where our CIS-guided timing strategy is compared to the strong baselines NTI+P2P and Stable Flow using CLIP-based metrics that are common in the field. These results show that PCI’s temporal insights lead to more controlled and semantically aligned edits, even without modifying the underlying architecture.
>
> | **Method**          | $\mathbf{CLIP}_{img}$ | $\mathbf{CLIP}_{txt}$ | $\mathbf{CLIP}_{dir}$ |
> |---------------------|--------------|--------------|--------------|
> | NTI+P2P             | 0.8666       | 0.2215       | 0.0979       |
> | Stable Flow (CVPR-2025)         | 0.8324   | 0.2152       | 0.0631       |
> | **PCI-τ$_{30}$**         | 0.9343       | 0.2125       | 0.1014       |
> | **PCI-τ$_{50}$**         | **0.8885**   | 0.2236       | 0.1387       |
> | **PCI-τ$_{60}$**         | 0.8625       | 0.2289       | 0.1531       |
> | **PCI-τ$_{70}$**         | 0.8353       | **0.2341**   | **0.1678**   |
> | **PCI-τ$_{90}$**         | 0.7679       | 0.2449       | 0.1963       |
>
> ---
> **References**
>
> [1] O. Avrahami, O. Patashnik, O. Fried, E. Nemchinov, K. Aberman, D. Lischinski, and D. Cohen-Or, “Stable Flow: Vital Layers for Training-Free Image Editing,” in *Proc. IEEE/CVF Conf. on Computer Vision and Pattern Recognition (CVPR)*, 2025.
>
> [2] R. Mokady, A. Hertz, K. Aberman, Y. Pritch, and D. Cohen-Or, “Null-Text Inversion for Editing Real Images using Guided Diffusion Models,” in *Proc. IEEE/CVF Conf. on Computer Vision and Pattern Recognition (CVPR)*, 2023.
>
> [3] Y. Chen, Y.-X. Wang, D.-A. Huang, L.-J. Li, R. Krishna, and Y. Xu, “LLaVA-OneVision: Easy Visual Task Transfer,” *arXiv preprint arXiv:2407.13531*, 2024.
>
> [4] T. A. Adediran and S. Al-Dabagh, “SmolVLM: Redefining small and efficient multimodal models,” *arXiv preprint arXiv:2405.09334*, 2024.
>
> [5] J. Chen et al., “PixArt-α: Fast Training of Diffusion Transformer for Photorealistic Text-to-Image Synthesis,” *arXiv preprint arXiv:2310.00426*, 2023.
>
> [6] Black Forest Labs, “FLUX,” 2024. Online: https://github.com/black-forest-labs/flux

---

### Author Response · Authors · 2025-11-21
**Author response summary**

We deeply appreciate the reviewers’ detailed and constructive evaluations, their supportive comments, and their clear appreciation of the strengths of our work. We are grateful that across reviewers, our paper was praised for its clear writing, well-presented figures, and self-contained math (RMvL, uBmP, V3XN). The core idea of PCI was described as intuitive, straightforward, and well-motivated (RMvL, V3XN), with Qc2X highlighting its training-free, model-agnostic nature and effective seed-resampling strategy. Reviewers also emphasized the novelty and value of analyzing the temporal evolution of concepts in diffusion models (Qc2X, uBmP). We greatly appreciate these positive assessments.
During the rebuttal, we made substantial improvements inspired by all reviewers’ constructive feedback. In particular, we:

**Expanded the empirical scope and robustness of PCI/CIS.**
- Added two additional generative models: **PixArt-α** (diffusion, DiT-based) and **FLUX.1-dev** (rectified flow),showing that temporal concept behaviors generalize across UNet, DiT, and rectified-flow architectures.
- Added two additional VQA judges: **LLaVA-OneVision-7B** and **SmolVLM-2B**, and reported multi-judge CIS curves, demonstrating strong consistency across four LVLMs.

**Strengthened the evaluation of editing quality.**
- Introduced a **large-scale quantitative editing evaluation** on **1,760 images** across **88 concepts**, comparing CIS-guided PCI to **NTI+P2P** and **Stable Flow** using three CLIP-based metrics.
- Demonstrated that the CIS-guided window **[0.5, 0.7]** yields the best balance of insertion strength and content preservation, directly addressing concerns about Fig. 5.
- Added further qualitative comparisons to baselines and clarified the visual presentation of editing results.

**Extended PCI beyond single-concept insertion.**
- Added a **concept deletion analysis (CDS)** to study when concepts become impossible to remove, offering a complementary “disappearance” perspective.
- Added **multi-concept interaction experiments**, revealing compositional cases and behaviors (e.g., *female + sketch*).
- Added analysis of **abstract concepts** (e.g., *elegant*, *art*, *creative*) and discussed VQA limitations and mitigation strategies.

**Improved robustness analyses.**
- Clarified in the main text that CIS curves are **averaged over many seeds** and added an ablation identifying how many seeds are needed for stable estimation.
- Added a **prompt-variation robustness test**, showing that temporal concept ordering is preserved across paraphrased prompts.

**Added qualitative interpretability tools.**
- Included a **cross-attention analysis** that visualizes how spatial focus and concept strength evolve across intervention timesteps, complementing the binary CIS metric with a continuous, spatial perspective.

**Corrected presentation issues.**
- Fixed all citation formatting issues and improved clarity in several sections as suggested by the reviewers.

We thank the reviewers again for their constructive feedbacks, which directly contributed to a significantly stronger and more comprehensive paper.

---

> ### Author Response · Authors · 2025-12-02
> **Author response summary extension**
>
> We have added a brief extension to our official author response summary to ensure that the key updates made during the rebuttal are clearly visible at a glance, including the broader framing around temporal concept dynamics, the reorganized appendix, and the expanded multi-model analysis. We sincerely thank the ACs for their time and high-effort throughout the review process.
>
> **From CIS to broader temporal concept dynamics.**
>
> We further clarified that our main contribution is a general framework for temporal concept dynamics via PCI, of which CIS (insertion) and CDS (deletion) are two complementary instantiations. In support of this, we followed the reviewer's suggestions and added a dedicated CDS analysis to the Appendix to study when concepts become impossible to remove, and reorganized the Appendix so that CIS and CDS are clearly separated and easier to navigate. Following Reviewer uBmP’s suggestion, we also updated the title to the more general
> “Temporal Concept Dynamics in Diffusion Models via Prompt-Conditioned Interventions”,
> and revised the introduction and method sections to better reflect this broader perspective that our additional experiments now provide.
>
> **Additional clarification and analysis.**
>
> Following Reviewer Qc2X, we clarified that regarding our multi-concept PCI experiments (e.g., female + Asian, coat + snowy, female + sketch), the CIS is a concept-specific metric even under multi-concept prompts. This reveals both near-compositional cases and non-linear interactions where joint insertion reshapes temporal insertion windows. For abstract concepts (art, creative, elegant), we further extended the analysis across four LVLMs, showing that larger VLMs produce closely aligned CIS trajectories even for high-level concepts.

---

### Meta-Review · Area_Chair_aGtK · 2026-01-19

**Summary:**

Reviewers agreed the paper is clearly written and that PCI/CIS is an intuitive, training-free way to probe when prompt concepts still affect generation, but several raised concerns about interpretation stating that CIS measures the latest timestep a concept can be inserted and does not directly show when the model first encodes a concept (from rev V3XN). Also validity mainly concern about dependence on LVLM/VQA and a binary present/absent score could bias CIS curves (this is mostly from revs uBmP, RMvL).
On practical significance, the point are about editing evidence initially looked qualitative with limited baselines and no quantified insertion–preservation trade-off (uBmP, Qc2X, RMvL); and finally they stated that results were initially tied to SD-series and simple prompts (RMvL, Qc2X). The rebuttal directly addressed most of these concrete issues by adding additional generative models (PixArt-$aplha$, FLUX.1-dev), multiple VQA judges, seed/prompt robustness tests, multi-concept and deletion analyses, and a large-scale quantitative editing evaluation with baselines, leaving the remaining limitations mainly about how strongly the temporal curves can be interpreted beyond intervention-based controllability.

Overall discussions lean toward supporting a weak accept.

**Reviewer Concerns:**

The rebuttal substantially addressed the key concerns about scope and evidence by adding more generative models (beyond SD-only), multiple LVLM/VQA judges, robustness checks over seeds and prompt paraphrases, and a quantitative editing study with baselines that measures the insertion–preservation trade-off. The main remaining issue is interpretation: PCI/CIS (and the added deletion analysis) still primarily measure intervention-based controllability/irreversibility, not when a concept is first internally encoded, and binary VQA limits (strength/binding/localization) are only partially mitigated.

**Reviewer Scores:**

- Rev RMvL could likely up after added non-SD models, multi-judge evaluation, and quantitative editing results, though still possibly cautious on novelty.
- Rev V3XN slightly up but most likely remain the same given added robustness analyses; core interpretation caveat remains.
- Rev Qc2X likely same, score is already high. The rebuttal directly addresses preservation trade-off and compositionality concerns.
- Rev uBmP could go slightly up with quantified editing + baselines, multi-judge agreement, broader scope, and improved framing/organization.

These are my most optimistic guess

---

### Decision · Program_Chairs · 2026-01-26

Accept (Poster)